# The Arabidopsis AtSWEET13 transporter discriminates sugars by selective facial and positional substrate recognition
Austin T. Weigle [1] & Diwakar Shukla [2,3,4,5] ✉

Transporters are targeted by endogenous metabolites and exogenous molecules to reach cellular destinations, but it is generally not understood how different substrate classes exploit the same transporter's mechanism. Any disclosure of plasticity in transporter mechanism when treated with different substrates becomes critical for developing general selectivity principles in membrane transport catalysis. Using extensive molecular dynamics simulations with an enhanced sampling approach, we select the *Arabidopsis* sugar transporter AtSWEET13 as a model system to identify the basis for glucose versus sucrose molecular recognition and transport. Here we find that AtSWEET13 chemical selectivity originates from a conserved substrate facial selectivity demonstrated when committing alternate access, despite mono-/di-saccharides experiencing differing degrees of conformational and positional freedom throughout other stages of transport. However, substrate interactions with structural hallmarks associated with known functional annotations can help reinforce selective preferences in molecular transport.

Membrane transporters predominantly control the developmental and metabolic fates of cells through water, ion, and carbon allocation. Transport catalysis does not involve breaking or making covalent bonds to form some product; instead, membrane transporters undergo large conformational changes to simply change a substrate's physical position rather than its chemistry[1,2]. Whether on accident or by design, transporters uptake and export molecules outside of their known or canonical substrate scope[3]. Still, outside of just a few static crystal structures captured in different conformations, it is unknown exactly how a typical membrane transporter's alternate access mechanism can be generalized when discriminating between different substrates.

A best-case scenario for approaching this research question would be to involve a relevant model system that transports similar yet structurally distinct substrates and does so without simultaneous cotransport of ions or additional molecules. Surprisingly, there exist few cofactor-independent transporters with readily available crystal structures[4]. Sugars *W*ill *E*ventually be *E*xported *T*ransporters (SWEETs) are universally expressed, cofactor-independent, bidirectional uniporters responsible for sugar transport in plants[5–7]. Evolved from bacterial SemiSWEETs possessing a single triple helix bundle, plant SWEETs contain two triple helix bundles connected by

an inverted transmembrane helix linker (TM4)[8,9]. The evolutionary advent of TM4 introduced topological pseudo-symmetry through inverted repeats, likely to structurally reinforce the functional need for bidirectional transport activity[10]. SWEET substrate specificity generally aligns with sequence similarity: phylogenetic clades I, II, and IV transport monosaccharides, while clade III prefers sucrose but has also demonstrated glucose and gibberellin transport activity[11–15]. Their tissue-based expression and substrate specificities for mono- and di-saccharides underscore their significance in maintaining plant physiology through proper sugar signaling. This signaling can be disrupted during plant-pathogen interactions, where pathogens hijack SWEET functionality to redirect sugars away from the plant host. Specifically, clade III SWEET promoters are targeted during *Xanthomonas* infection to alter their expression and facilitate bacterial blight pathogenesis —an etiology currently known to devastate rice yields for African and Asian subsistence farmers[11,16,17].

Given its substrate scope (glucose, sucrose, and gibberellin), cofactor independence, physiological/societal importance as a clade III SWEET, and available crystal structure (PDB ID: 5XPD)[18], AtSWEET13 emerges as an excellent candidate for studying how different substrates activate the same membrane transporter. Beginning with the inward-facing (IF) AtSWEET13

[1]Department of Chemistry, University of Illinois at Urbana-Champaign, Urbana, IL 61801, USA. [2]Department of Chemical & Biomolecular Engineering, University of Illinois at Urbana-Champaign, Urbana, IL 61801, USA. [3]Department of Plant Biology, University of Illinois at Urbana-Champaign, Urbana, IL 61801, USA. [4]Department of Bioengineering, University of Illinois at Urbana-Champaign, Urbana, IL 61801, USA. [5]Center for Biophysics and Computational Biology, University of Illinois at Urbana-Champaign, Urbana, IL 61801, USA. ✉e-mail: diwakar@illinois.edu

crystal structure[18], we implemented an adaptive-sampling-based regime with classical molecular dynamics (MD) simulations. Similar approaches have proven to be excellent strategies for studying membrane transport protein mechanisms[19–23]. Herein, we resolve the conformational landscapes depicting AtSWEET13 *apo*, *holo* glucose (GLC), and *holo* sucrose (SUC) transport cycles. After statistically validating our ~450 μs of aggregate simulation with Markov State Models (MSMs)[24–28], we identified regions along the AtSWEET13 transmembrane channel critical for differentiating GLC from SUC molecular recognition.

## Results

### Resolving the AtSWEET13 transport cycle

AtSWEET13 gating dynamics can be described by the extracellular and intracellular gating distances between Lys65–Asp189 and Phe43–Phe164, respectively[18]. In line with prior work on SWEET gating dynamics[19,20], the inward-facing (IF) conformational state occupies the lowest free energy state (Fig. 1a). This finding corroborates SWEET dependence on biological sugar

concentration gradients[5,11], as this bidirectional uniporter family can be expected to remain in the IF state until enough sugar has accumulated to require export to sink destinations. Additionally, the MSM-weighted *apo* AtSWEET13 transport cycle includes an hourglass-like (HG) intermediate state, which experiences a higher energy barrier for transitioning directly to the outward-facing (OF) state rather than through the occluded (OC) state (Fig. 1b). HG states have been previously observed during the transport cycle of rice glucose transporter OsSWEET2b, where the HG conformation preserves the crux of alternate access by constricting the transmembrane channel pore radius[19]. Meanwhile, AtSWEET13 conformational change experiences higher energy barriers for a transition during *holo* GLC and SUC transport cycles, where access to intermediate states in GLC transport are ~0.9–1.6 ± 0.7 kcal mol$^{-1}$ higher, and ~0.2–0.3 ± 0.3 kcal mol$^{-1}$ higher in SUC transport, when compared to *apo*. Like the *apo* simulations, both MSM-weighted *holo* datasets indicate the IF conformation as the lowest energy state; however, *holo* simulations uniquely stabilize transitions for an alternate access mechanism passing through an HG conformation while

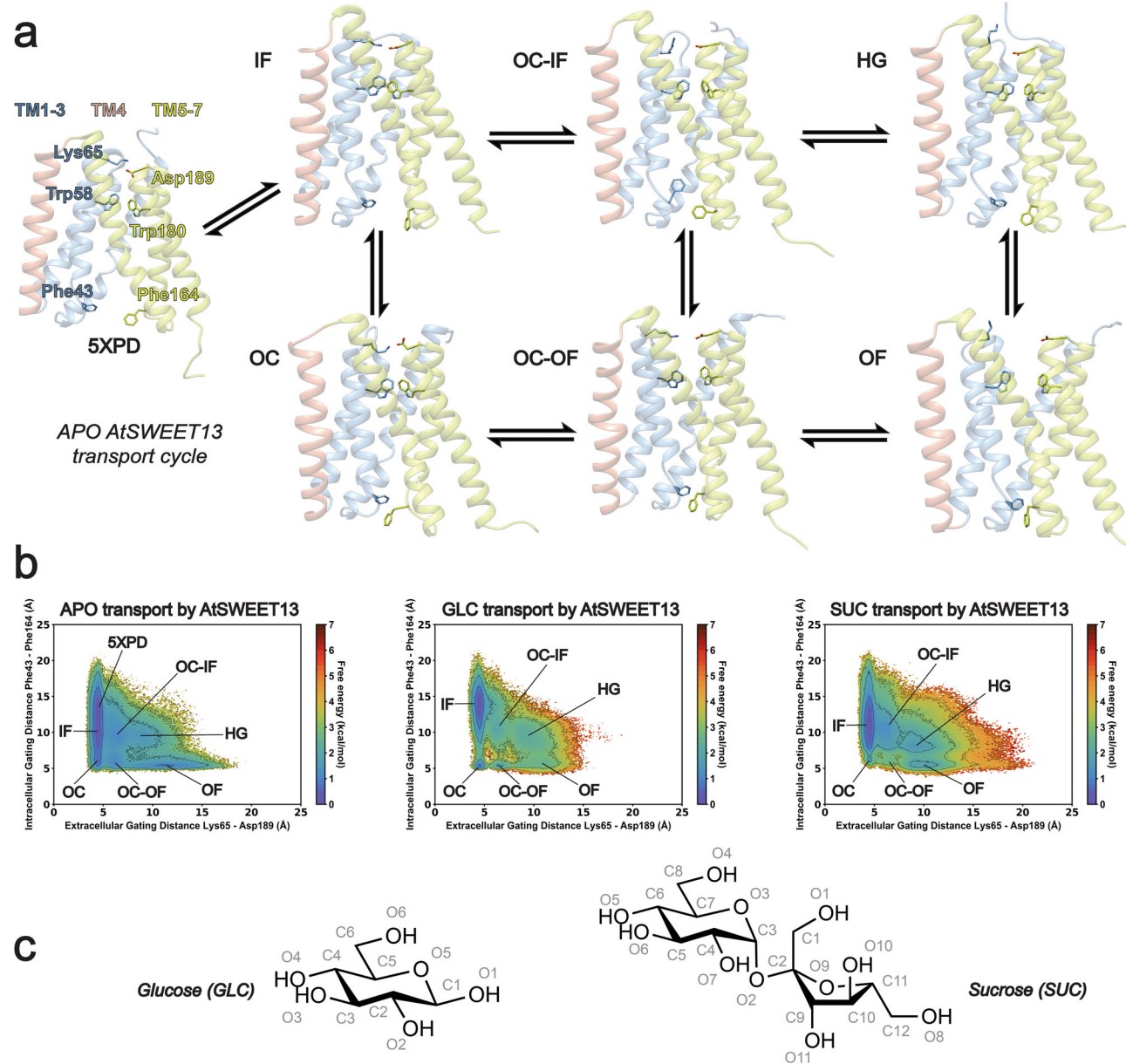

**Fig. 1 | AtSWEET13 transport cycle. a** *apo* conformational snapshots for each representative state. **b** Gating landscapes for *apo*, GLC, and SUC transport processes, with key intermediate states labeled based on extracellular and intracellular gating

distances. **c** Glucose (GLC) and sucrose (SUC) substrate molecular structures with atoms numbered as parameterized for molecular dynamics simulations.

destabilizing an OC-based alternate access pathway. An HG gating distance of ~8–10 Å, rather than an OC aperture of ~5 Å, is likely needed to accommodate the molecular recognition processes for each substrate (Fig. 1c). It is important to note that the existence of HG states in *holo* sugar transport does not violate the principle of alternate access, as this common transporter topology maintains a blocked channel to prevent substrate leakage[19,29,30]. Overall, these findings are consistent with the "free ride" mechanism, where the *apo* and *holo* transport cycles access the same protein conformational changes, albeit at different relative free energies and transition rates[31]. The relationship between the higher energy barriers for GLC-induced gating versus SUC-induced gating also corroborates experimental evidence that AtSWEET13 has evolved to be a more efficient disaccharide transporter[18].

## Alternate access for selective sugar transport is distinguished by altered energy barriers and distinct residue fluctuations

To determine exactly how AtSWEET13 differentiates GLC versus SUC transport, free energy landscapes specific to individual substrate atoms were projected versus substrate position and AtSWEET13 gating distances (see the "Methods" section). Figure 2 depicts the gating difference versus substrate atom position landscapes, where the atom under inspection is the hexose backbone ether oxygen (Fig. 2a, b). The hexose backbone ether oxygen (GLC O5, SUC O3) is selected because one would initially assume these atoms to be representative of hexose transport within monomeric GLC or the pyranose ring in SUC. Although sugar pyranose rings can have variable chiral preferences for their alcohol groups, characteristics of this backbone ether oxygen are otherwise invariant.

Upon binding from the intracellular face of the transporter, both sugars must traverse 25 Å up into the transmembrane channel to reach the hydrophobic binding pocket containing Trp58 and Trp180. Due to the topological asymmetry of AtSWEET13, where the hydrophobic binding pocket containing Trp58 and Trp180 is arranged more closely to the OF-side of the transmembrane channel, much of GLC and SUC translocation occurs while AtSWEET13 is in an IF-favoring conformation (Fig. 2a, b). Indeed, as the sugars approach the Trp58–Trp180 binding pocket, the intracellular gating distance is ~10 Å greater than the extracellular gating distance for GLC and ~7 Å greater for SUC. Binding events along the IF association pathway for SUC ($SUC_1 \rightarrow SUC_2 \rightarrow SUC_3$) are at least ~1.0 ± 0.1 kcal mol$^{-1}$ more stable than as seen for GLC ($GLC_1 \rightarrow GLC_2$; Fig. 2a, b). Both GLC and SUC then approach a stable intermediate state after climbing 25 Å up the transmembrane channel ($GLC_3$; $SUC_5$), during which point AtSWEET13 commits each substrate to alternate access for the remaining 15 Å of membrane-spanning distance. Projection of raw gating distance data onto these gating difference distance plots reaffirms how alternate access occurs through an HG pathway (Supplementary Figs. 1 and 2). Commitment to alternate access is reflected based on the conformational changes between the $GLC_3 \rightarrow GLC_4$ states and the $SUC_5 \rightarrow SUC_6$ states, where $SUC_6$ is ~1.2 ± 0.3 kcal mol$^{-1}$ more stable than $GLC_4$ (Fig. 2a, b). These transitions invoke the greatest protein conformational change, where the per-residue RMSD of nonterminal amino acids is surprisingly greater for the GLC HG-OF transition than the same transition for the transport of the larger SUC molecule (Fig. 2c, d).

Much of the same conformational change by AtSWEET13 occurs during both GLC and SUC transport (Figs. 2c–f and 3). TM1–TM2 loop linker residues Arg33–Pro47 demonstrate higher fluctuations during GLC and SUC transport. Particularly, Arg33 reaches RMSD values between ~4.0 and 6.5 Å during initial IF-binding and alternate access events. Glu41, Cys42, and intracellular gate residue Phe43 consistently reach ~7 Å RMSD values during SUC IF binding and GLC alternate access. TM2–TM3 loop linker and adjacent residues Ala62–Phe78 fluctuations are largely involved in the commitment to alternate access. Maximum RMSD values for Thr68 are 6.47 Å for GLC and 7.37 Å for SUC transport events, respectively. These TM2–TM3 residues also include extracellular gate residue Lys65, which has an RMSD of ~4 Å for both sugar transport processes during alternate access. Fluctuations for inverted transmembrane helix linker TM4 Phe90–Leu112

are also involved during alternate access, although Phe111 is implicated in initial IF binding events with RMSD values of 4.22 Å for GLC transport and 4.79 Å for SUC transport. TM4–TM5 loop linker residues Lys126–Lys132 demonstrate their fluctuations during alternate access, while Lys126 fluctuations are also characteristic of IF-binding events for both transport processes. TM5 Phe141–Met165 experiences the greatest fluctuations during IF-binding events for SUC transport but predominantly during alternate access events for GLC transport. Still, Phe141–Met165 fluctuations during IF-binding events for GLC transport are also high. Among this TM5 selection, some of the highest RMSD values are seen for Arg158, Thr159, Arg160, Glu163, Met165, and intracellular gate residue Phe164. Lastly, residues Leu185–Phe190 show larger RMSD values during commitment to alternate access; this includes extracellular gate residue Asp189.

Noteworthy of these highly fluctuating regions is that the residues involved in the largest extent of conformational change do not reside within the central binding pocket. These residues instead predominantly line the AtSWEET13 periphery, echoing the importance of distant residues for evolved function in other proteins, including membrane transporters[32–35]. From the perspective of transporter structure, these residues occupy regions directly associated with gating dynamics and potential gating contacts. Still, AtSWEET13 maintains high fluctuations within these regions for both GLC and SUC transport reactions, suggesting that movements from these residues are essential for general alternate access regardless of substrate.

Most of the larger RMSD deviations are incited as AtSWEET13 commits to alternate access, making residue fluctuations associated with this exact conformational change a good instance for comparing relative protein dynamics in response to sugar transport. All averaged per-residue RMSD values across the metastable states for each sugar transport process were compared against the alternate access state for the opposite sugar transport process (i.e., states $GLC_4$ and $SUC_6$ in Fig. 2; Supplementary Fig. 3). A suite of residues with statistically significant differences in RMSD was identified, highlighting how the GLC (Fig. 3a) and SUC (Fig. 3b) alternate access schemes distinctly differ from the entirety of each opposite transport process.

Residues with statistically significant differences between the GLC and SUC alternate access schemes surround the extracellular vestibule near and above the conserved Trp58–Trp180 binding pocket, as well as the intracellular facing portion of TM4. Among all sites, the greatest differences exist between the TM2–TM3 loop linker and adjacent residues, which constitute the triple helix bundle 1 extracellular gate. These residues help maintain an IF-like state throughout the entirety of transport events due to the topological asymmetry of AtSWEET13, as well as directly participate in extracellular vestibule widening during HG to OF transitions. In total, these RMSD analyses help explain what residue-specific dynamics deviate throughout conformational change associated with GLC versus SUC transport events.

## Evaluating the contribution of conformational change versus ligand recognition to explain differences in AtSWEET13 sugar transport

RMSD analyses from the previous "Results" section helped determine what parts of AtSWEET13 respond and fluctuate differently during GLC versus SUC transport. However, selective differences between sugar transport events must also exist on the basis of ligand recognition. Each of the heavy atoms within GLC present similar gating landscapes (Supplementary Figs. 4 and 5). When projecting the transport data as gating difference versus atom position using each heavy atom in SUC as the reference substrate atom, the equivalent of the $SUC_5 \rightarrow SUC_6$ transition (where the substrate atom $Z$ position is ~5 Å) is smoothened for fructosyl atoms (Supplementary Figs. 6 and 7). Meanwhile, projecting transport landscapes with SUC glucosyl atoms show a persistent free energy barrier at the exact same stage of transport, just as was seen with respect to the transport of monomeric GLC (Supplementary Figs. 8 and 9). AtSWEET13's preference for SUC transport thus appears to be reflected by the evolved incorporation of the pentose moiety into the disaccharide structure, where its presence helps facilitate

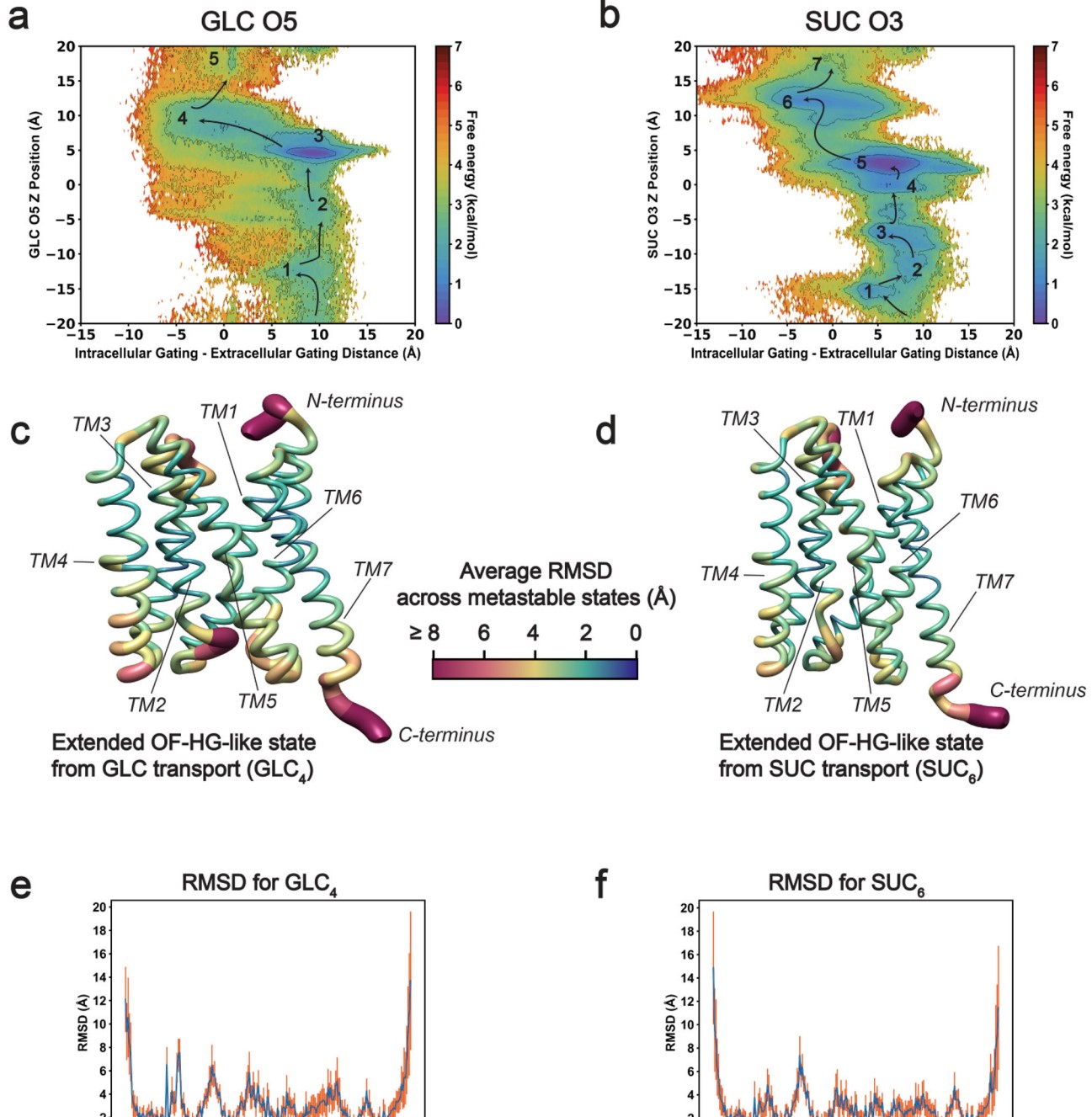

**Fig. 2 | Protein conformational change implicated in sugar transport. a** GLC transport versus the difference in intracellular versus extracellular gating. AtSWEET13 is in a more IF-like position, the more positive the gating difference, and a more OF-like position, the more negative the gating difference. **b** SUC transport versus the difference in intracellular versus extracellular gating. **c** AtSWEET13 OF-HG-like conformation immediately following commitment to alternate access in GLC transport. **d** AtSWEET13 OF-HG-like conformation immediately following commitment to alternate access in SUC transport. **e** RMSD values relating to residue-specific fluctuations in response to alternate access during GLC transport over 157 independent frames from the MSM-weighted minima. **f** RMSD values relating to residue-specific fluctuations in response to alternate access during SUC transport. Arrows drawn over landscapes are for illustrative purposes to suggest the transport path according to the lowest MSM-weighted free energy transitions as shown. RMSD values are calculated as an average in comparison to the number of states (simulation frames) found at the center of each energetic minima characterizing each enumerated state in panels (**a**) and (**b**). RMSD error bars are reported as the average ± SD. $N = 134,392$ pairwise simulation frame RMSD calculations in (**e**) and $N = 743,273$ pairwise simulation frame RMSD calculations in (**f**).

commitment to alternate access better than the SUC hexose unit. Additionally, SUC atom positioning throughout the early IF uptake stages of transport does appear to be more stable than seen for GLC. Aside from the differential participation of residues determined from RMSD analyses, these results suggest that intracellular transport binding stages apart from the HG

states are controlled by different processes for the translocation of the two sugars.

Time-lagged independent component analysis (tICA) used to determine appropriate features for MSM construction and validation further corroborates how GLC transport is more reliant on protein conformational

**Fig. 3 | Protein conformational change implicated in sugar transport. a** RMSD fluctuations throughout GLC transport that are statistically significantly different from RMSD fluctuations seen during AtSWEET13 alternate access for SUC transport. **b** RMSD fluctuations throughout SUC transport are statistically significantly different from RMSD fluctuations seen during AtSWEET13 alternate access for GLC transport. The color bar represents the number of states for which the RMSD values significantly differ from that of the alternate access state of the opposite ligand transport process. HG-like states undergoing alternate access are shown as snapshots for GLC (left) and SUC (right) transport, respectively. Residues with heatmap entries marked with an asterisk showed statistical significance at $p < 0.01$ by a two-sided $t$-test. Residues without an asterisk showed statistical significance at $p < 0.05$.

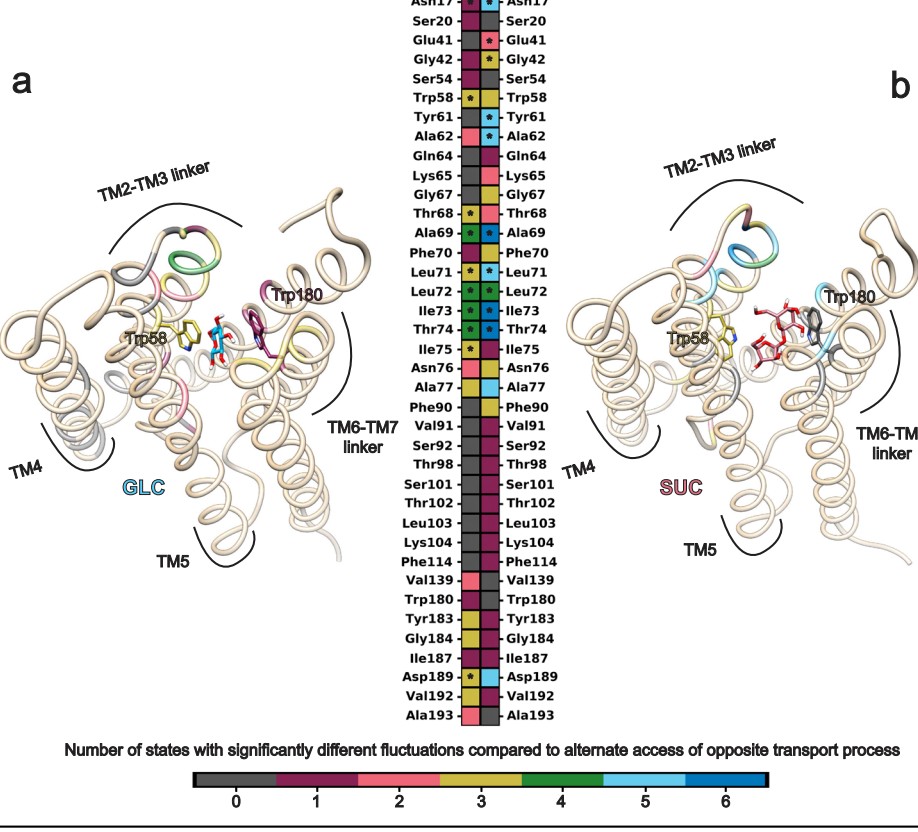

change (Supplementary Fig. 1; Supplementary Table 1)[36,37]. Second-order tICA decomposition has demonstrated that the first and second-time-lagged independent components (tICs) relate to the slowest, or rate-determining, processes for a given biomolecular system[36]. During initial iterations of MD data featurization, exhaustive attempts were made to ensure that the amount of adaptive sampling carried out sufficiently captured the dynamics of transport. To this end, the aggregate datasets for *apo*, GLC, and SUC transport were described using between 100 and 1000 residue-residue distances for transmembrane channel residue pairings, along with substrate atom-residue distances to the select channel residues (see the "Methods" section).

Representations of these more feature-diverse tICA discretization attempts for AtSWEET13 transport were used to shortlist a set of features that would best describe transport dynamics contained within the aggregate dataset in a final MSM (Supplementary Fig. 10). An appropriate description of transport involves features that properly discretize metastable transport states from one another. Additionally, resulting discretization should not cause free energy landscape reweighting that artificially deviates from trends seen during raw data collection based on poor clustering. Iterative attempts at MSM construction were made until feature sets satisfied these criteria (see the "Methods" section).

After feature set reduction, the slowest tICs for monosaccharide transport were highly correlated with 13 protein residue-residue distances, whereas just six residue-residue distances were highly correlated with disaccharide transport (Supplementary Tables 2 and 3). Additionally, tICA landscape projection depicts *apo* and SUC transport as dominated by a singular process while GLC transport is highly correlated along two distinct processes (Supplementary Fig. 1). For *apo* transport, $tIC1^{apo}$ signified protein conformational change (Supplementary Table 4), as the most correlated features related to protein residue-residue distances. For SUC transport, $tIC1^{SUC}$ reflected glucosyl moiety atom positioning while $tIC2^{SUC}$ reflected fructosyl moiety atom positioning (Supplementary Tables 3 and 5), together suggesting overall substrate translocation is rate-limiting. Lastly, for GLC transport, $tIC1^{GLC}$ reflected protein conformational change while $tIC2^{GLC}$

reflected monosaccharide substrate atom positioning (Supplementary Tables 2 and 6), suggesting greater extents of protein conformational change are required for GLC transport (see the "Methods" section concerning MSM construction and feature selection).

Despite the increased dependence on protein conformational change for monosaccharide transport, transmembrane channel pore radius between GLC and SUC transport share similar size as well as morphological changes during the transport cycle (Supplementary Figs. 3 and 11). AtSWEET13 exhibits a similar extent of IF gating aperture when presented with monosaccharide GLC versus disaccharide SUC, despite SUC being a larger molecule. During transport of either substrate, major changes in transmembrane channel pore radius only occur after commitment to alternate access (Supplementary Fig. 1). From the perspective of protein conformational change, these findings signify that AtSWEET13 surveys a relatively similar distribution of protein dynamics to position each native substrate into the Trp58–Trp180 binding pocket before committing each to alternate access. Since the general gating mechanics of AtSWEET13 are predicted to remain relatively the same between the two substrate transport processes, it is more likely that preferential substrate-specific interactions occurring within the transmembrane channel may act as determinants of selectivity.

## Stages of discriminative sugar recognition occur across distinct segments of transmembrane-spanning channel residues

The extent of similarity in AtSWEET13 gating is somewhat expected, as one protein can only be so plastic to maintain the transport of different classes of molecules. However, the differences in how energetically favorable intermediate states were along IF association pathways shown in Fig. 2 are intriguing. GLC O5 and SUC O3 represent nearly identical backbone hexose ether oxygen atoms, so one would initially assume that these atoms yield similar free energy landscapes. Conversely, atoms along SUC pentose atoms experienced lower energy barriers during commitment to alternate access (Supplementary Figs. 5 and 6). If the incorporation of a glycosidic linkage is enough to make seemingly chemically equivalent atoms between GLC and SUC no longer perceived as functionally equivalent by AtSWEET13, and if

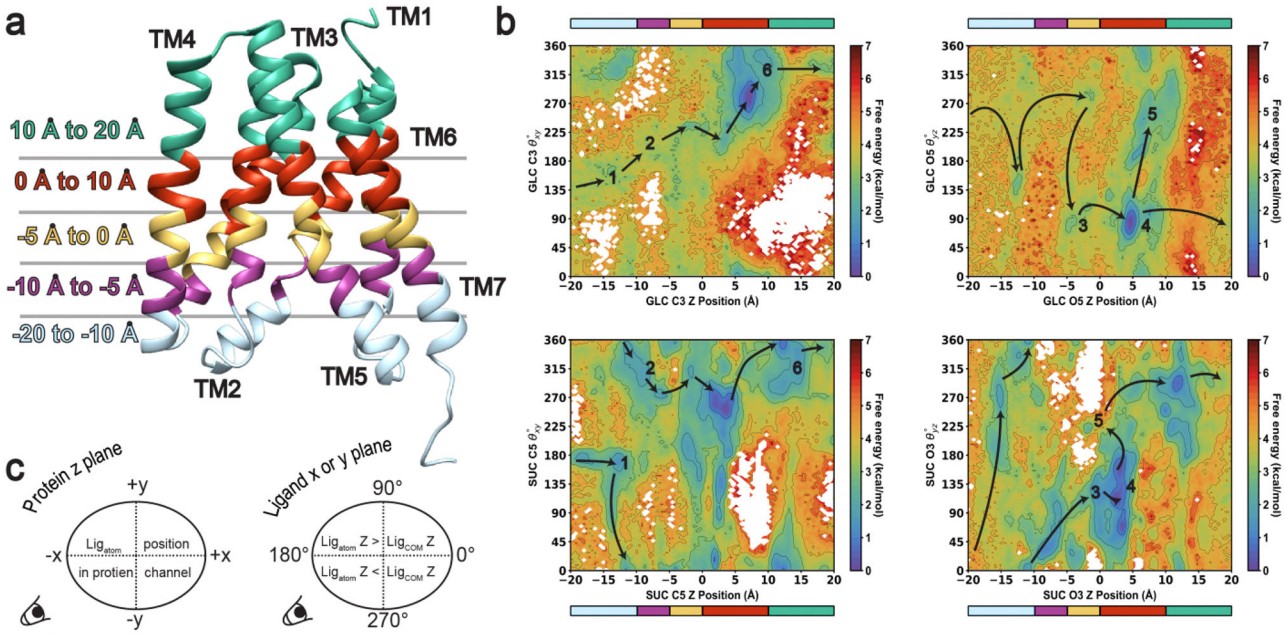

**Fig. 4 | AtSWEET13 layers for discriminating sugar substrates. a** AtSWEET13 5XPD crystal structure with layered coloring based on residue roles in the stage of sugar transport. **b** Atom-specific **θ** rotational plots, measuring ligand atom Z position versus rotational orientation. **θ**$_{xy}$ is shown between nearly equivalent atoms GLC C3 and SUC C5, while **θ**$_{yz}$ is shown for GLC O5 and SUC O3. Colorbars above and below each plot allow direct comparison to the colored 5XPD structure in (**a**). **c** Schematic explanation for interpreting **θ**$_{xy}$ and **θ**$_{xz \, or \, yz}$ calculation. **θ**$_{xy}$ captures the angle between the protein center of mass (COM) and the specific ligand atom coordinates in the XY planes. **θ**$_{xz \, or \, yz}$ captures the angle between the ligand (COM) and the specific ligand atom coordinates in the XZ or YZ planes. Arrows drawn over landscapes are for illustrative purposes to suggest the transport path according to the lowest MSM-weighted free energy transitions as shown.

the two monomeric units of SUC experience different energy barriers, then atom-specific molecular recognition may be satisfactory in explaining how the two sugar types are differentiated.

Treating AtSWEET13 as a "physical enzyme"[1,2] requires taking facial and spatial selectivity into consideration. Accordingly, each GLC and SUC atom was analyzed via a vector representation with respect to its Z position during transport (see the "Methods" section for AtSWEET13 atom-specific analyses). Briefly, the angle between the atom of interest and the center of mass of the protein along the XY plane (**θ**$_{xy}$) indicates ligand spinning or spatial selectivity. Conversely, the angle between the atom of interest and the ligand's center of mass along the XZ or YZ planes (**θ**$_{xz}$ or **θ**$_{yz}$) indicates ligand flipping or facial selectivity (Fig. 4c). Results for these analyses show that the perception of some atoms follows similar overall/inversion trends, but that the free energies for certain transitions differ even within the same sugar substrate.

Just like from our transport analyses in Fig. 2, we want to see how GLC and SUC transport differ by comparing what appear to be the most structurally and functionally similar atom groupings. Rotational analyses for each substrate-heavy atom are provided in the Supplementary Information (Supplementary Figs. 12–29). Definitive sugar functional groups are known to act as important handles for recognition by sugar transporters based on structure–activity relationships. In the case of mammalian GLUT transporters, GLUT1 shows equal hydrogen bonding preference to all but the GLC C2/O2 hydroxyl[38,39]; meanwhile, GLC recognition by GLUT2 and GLUT3 has been accredited to strong interactions with the C1/O1 hydroxyl[40]. Based upon our vector analyses, AtSWEET13 appears to recognize the GLC C3/O3 and SUC C5/O6, as well as the GLC C4/O4 and SUC C6/O5, hydroxyl groupings as roughly equivalent (Supplementary Fig. 30). The same C3/O3 and C4/O4 hydroxyls have been found to be most critical for glucose substrate recognition, as well as inhibitor development, for the malaria parasite transporter PfHT1[32,41]. Figure 4b presents the **θ**$_{xy}$ spinning rotational analyses for GLC C3 and SUC C5 for distinguishing GLC from SUC IF, as well as the **θ**$_{yz}$ flipping rotational analysis for GLC O5 and SUC O3. Most notably, discriminative binding events occur along

specific segments, or "layers", within the AtSWEET13 transmembrane-spanning channel (Fig. 4a). Between the **θ**$_{xy}$ and **θ**$_{yz}$ landscapes, we identify six different metastable states that correspond to different stages of sugar transport from the perspective of atom positioning.

Upon first binding to the IF AtSWEET13 state, GLC C3 and SUC C5 enter along the same face of the protein with $90° \leq$ **θ**$_{xy} \leq 200°$, or in between TM4 and TM5 (Fig. 4a, cyan layer; Supplementary Fig. 31, GLC$_1$/SUC$_1$). GLC IF association then must proceed within the range of $135° \leq$ **θ**$_{xy} \leq 225°$, as the GLC C3 atom samples transmembrane channel space between TMs 5–7 (Fig. 4a, purple later; Supplementary Fig. 31, GLC$_2$). Although averaging free energy of ~3.4 ± 0.3 kcal mol$^{-1}$, note that this is the only predicted IF association pathway available for GLC as it travels up the first 15 Å up the AtSWEET13 channel, inspiring us to refer to this rotational space as part of the "monosaccharide pathway" (Fig. 4b, top left panel, cyan and purple layers). SUC C5 can access the monosaccharide pathway during early IF association with similar free energy penalties as GLC C3 but can also access other regions of the channel at nearly half the free energy cost. SUC C5 instead is more likely to occupy regions where **θ**$_{xy} > 225°$ (Fig. 4b, bottom left panel, cyan, and purple layers). In contrast to GLC, the SUC$_2$ state, therefore, leads the GLC-like atom towards the TM6–TM1 cleft space, which the equivalent atom in the GLC structure does not approach during the same stage of transport. Thus, we term this area of AtSWEET13 as part of the "disaccharide pathway".

Regardless of mono- versus di-saccharide pathway for early IF association, the progression from an OC-IF to a more HG-like state results in experiencing similar rotational freedom. But like the gating difference plots from Fig. 2, SUC atoms experience lower energetic penalties as they sample **θ**$_{xy}$ space and contact each AtSWEET13 TM. One reason for this discrepancy in the energetic penalty in transit to the Trp58–Trp180 binding pocket is the varying degrees of facial freedom experienced by each substrate (**θ**$_{yz}$; Fig. 4b, right panels). During IF association, GLC can invert freely while occupying the TM5–TM6–TM7 cleft between the cyan and yellow AtSWEET13 layers (Fig. 4a). Interestingly, SUC atom IF association instead can only flip and cross the **θ**$_{yz} = 180°$ boundary within just the cyan layer. SUC

**Fig. 5 | Proposed mechanism of substrate recognition in AtSWEET13 transport.** Substrate inversion happens freely throughout GLC transport (**a**), making facial flipping not necessarily required for GLC to pass through the Trp58–Trp180 binding pocket during HG conformations and commitment to alternate access. For SUC transport (**b**), substrate inversion does not happen freely throughout the transport process, requiring substrate conformational rearrangement before AtSWEET13 can commit to alternate access. Figure inspired by the proposed mechanism for substrate binding in SUC1 by Bavnhøj, L. et al. (2023)[102]. Inspiration material from Bavnhøj, L. et al. is licensed under a Creative Commons Attribution 4.0 International License and permits adaptation (https://creativecommons.org/licenses/by/4.0/).

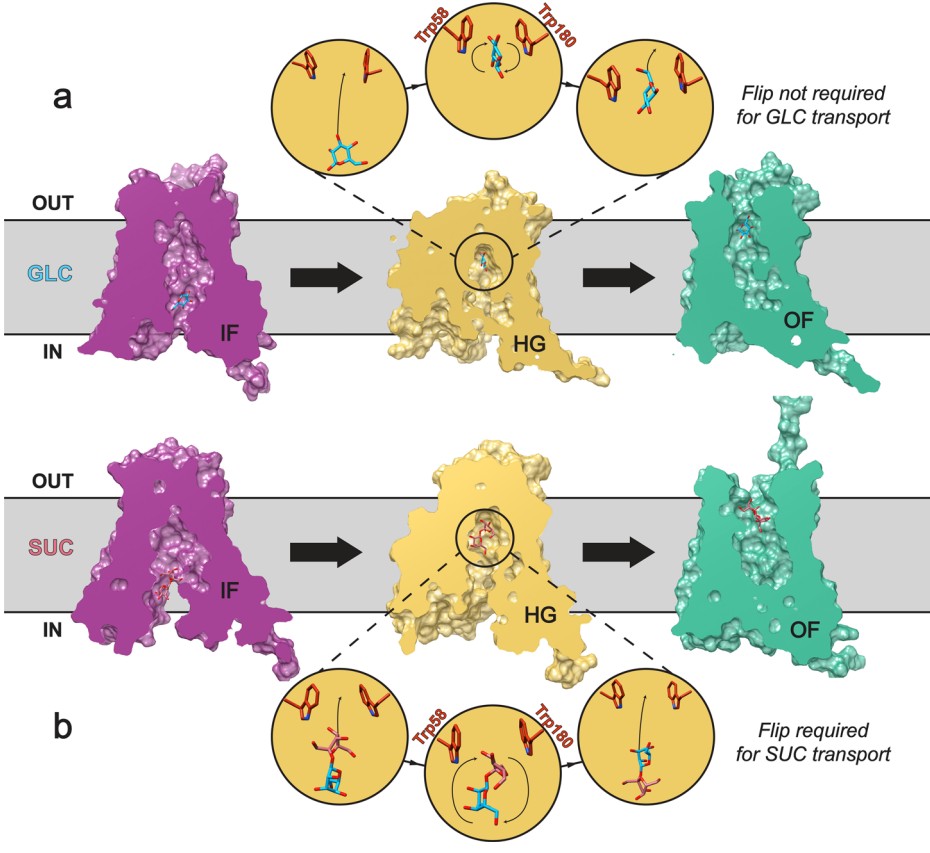

facial orientation is then relatively fixed when traveling throughout states $SUC_2 \rightarrow SUC_3 \rightarrow SUC_4$ (Fig. 4b, bottom right). On the other hand, GLC samples different facial orientations prior to binding to the Trp58–Trp180 pocket, where this conformational selection strategy is enabled by the smaller ligand's size. As SUC expresses less facial freedom, transport can rather focus on rotational substrate positioning such that a productive pre-transition state can be available for commitment to alternate access. Increased facial freedom for GLC instead results in variable orientations which likely require correction from more involved protein conformational change. AtSWEET13 is essentially playing a game of Rubik's Cube with the two substrates, where increased entropy in possible GLC facial orientations requires more correction by transmembrane-spanning channel residue interactions to enable a competent substrate orientation for transport catalysis.

Nevertheless, both predicted substrate transport pathways involve committing GLC and SUC to particular facial orientations to allow alternate access. Slab representation is used to highlight the role of substrate orientation more simply throughout the proposed mechanism for AtSWEET13 discriminative transport (Fig. 5; Supplementary Fig. 31 for an alternative view). SUC transport requires substrate inversion where the hexose unit is $\theta_{yz} < 180°$ and leads the disaccharide structure from the IF face of the transporter. Upon reaching the binding pocket, the pentose unit leads the disaccharide substrate towards OF dissociation as the hexose unit assumes $\theta_{yz} > 180°$ (Fig. 4a, b red layer; Fig. 5b, yellow HG state). Monosaccharide GLC can flip throughout what could be perceived as the transition state of translocation catalysis even while bound directly to Trp58 and Trp180 (Figs. 4b and 5a; $GLC_4 \rightarrow GLC_5$, Supplementary Fig. 31). Considering that the equivalent ether oxygens between GLC and SUC both have $\theta_{yz} \approx 225°$, it is possible that GLC flips freely during the transition state, perhaps signifying attempts to satisfy recognition requirements for either monomeric unit of SUC. Such an explanation is likely the primary reason why GLC samples the unproductive state $GLC_5$, as $GLC_3$ already expresses the correct facial orientation to commit directly from $GLC_3 \rightarrow GLC_4 \rightarrow GLC_6$ and final

OF dissociation. Substrate departure for GLC and SUC OF dissociation occurs through TMs 1 and 3 (Supplementary Fig. 31).

Overall, these atom-specific rotational analyses reveal that despite AtSWEET13 sampling the same conformational ensemble during GLC and SUC transport, the substrates experience different transport pathways. By considering transport from the perspective of the ligand, these analyses enable the identification of the structural portions within the AtSWEET13 transmembrane channel where the sugar transport pathways differ. These functional differences are also reflected in the perception of hexose and pentose units within SUC (Supplementary Figs. 12–29). Specifically, IF molecular recognition of the SUC hexose unit emphasizes that it can be perceived as monomeric GLC by potentially sampling the monosaccharide path. But this mode of transport for SUC is less probable based on MSM-weighted relative free energy calculations. AtSWEET13 specialization in disaccharide transport, therefore, emphasizes the importance of pentose molecular recognition during HG-OF transitions and OF dissociation.

### Describing distinct binding pathways for glucose versus sucrose sugar transport pathways

So far, our analyses suggest that the greatest distinguishing feature between mono- and di-saccharide transport by AtSWEET13 comes down to interactions between residue side chains and each transported sugar. Previous literature on protein molecular recognition of glycans suggests that dispersive van der Waals (vdW) contacts between sugar molecules and proteins are a dominant force in describing sugar binding[42]. This is mainly because the arrangement of C–H patches throughout sugar scaffolds creates ample opportunity for weak hydrophobic interactions between the carbohydrate structure and protein residues[43]. Outside of these multitudinous weak vdW interactions, sugar binding is anchored by particular strong hydrogen bonding and C–H···π stacking interactions[44]. Here, contact analyses were used to differentiate which AtSWEET13 residues interacted with the different sugar substrates the most during unbiased simulations (Supplementary Figs. 32 and 33).

Upon comparing the types of sugar-residue interactions observed among representative simulation frames, ~85% of GLC–protein interactions were due to vdW forces, and the remaining ~15% were mostly hydrogen bonds with protein residue side chains. SUC–protein interactions presented a similar ~88% of contacts based on vdW forces, while ~9% of contacts were from hydrogen bonds with residue side chains and ~3% were from hydrogen bonds with residue backbone atoms. In terms of specific interactions, AtSWEET13 follows literature trends in offering many aromatic residues for C–H⋯π stacking while simultaneously creating environments that can facilitate substrate repositioning through van der Waals contacts.

Sites for GLC early–IF association show strong sugar contacts with Ser151 and Gln44, as well as intracellular gating residue Phe43. Some C–H⋯π stacking can be observed with Phe43, helping to facilitate entry into the transmembrane channel from the solution (Supplementary Fig. 34a). Progression toward an extended HG-IF state has GLC maintains contacts with a triad of Pro47, Tyr48, and Glu83. C–H⋯π stacking does occur between GLC and Tyr48, although more dominant interactions happen with Glu83. Further translocation up the channel ultimately leads GLC towards extensive C–H⋯π stacking when bound to the Trp58–Trp180 binding site (Supplementary Fig. 34d). While within the Trp58–Trp180 binding site, GLC makes vdW contacts with nearby Leu72, Asn76, Phe146, and Asn196.

SUC binding shares some of the same contacts as GLC during initial IF binding. SUC engages in C–H⋯π stacking with intracellular gates Phe43 and Phe164, while also interacting with Gln44 and Ser151. The glucosyl moiety of SUC also becomes attracted to Pro47 and Tyr48, but the fructosyl moiety makes moderate contacts with many surrounding residues further up the transmembrane channel (Supplementary Fig. 35b). These moderate interactions become dominated by further C–H⋯π stacking between the glucosyl moiety and either Phe24 or Phe146, which were not as prevalent during GLC recognition. As SUC undergoes the facial inversion described in Figs. 4 and 5, Phe43 acts as an intracellular cap, maintaining a closed vestibule within the HG conformation so that alternate access can still proceed (Supplementary Fig. 35e). With its larger size, SUC then maintains a variety of C–H⋯π stacking interactions between Trp58, Trp180, and Tyr61, while also establishing polar contacts with other nearby binding pocket residues. Interestingly, the glucosyl moiety in SUC appears to make more consistent C–H⋯π stacking interactions between frames than monomeric GLC. Meanwhile, the fructosyl portion helps direct overall ligand positioning with its leading orientation prior to the necessary substrate inversion (Fig. 5).

Our identification of separate intracellular binding pathways highlights how a single SWEET transporter that transports different sugar types may demonstrate selectivity preferences. However, overall selectivity between members within a transporter family is likely to be associated with functionally divergent phylogenetic profiles. In the original work resolving the AtSWEET13 crystal structure, Han et al. used multiple sequence alignments to identify a subgroup of residues near the Trp58–Trp180 binding pocket whose patterns appeared to dictate substrate selectivity between different SWEET transporters[18,45]. Specifically, AtSWEET13 residues Val23, Ser54, Val145, and Ser176 constitute a "VSVS" motif common to disaccharide transporters. Meanwhile, solely monosaccharide-transporting SWEETs exhibit an "LNMN" motif along the same residues (i.e., Leu23, Asn54, Met145, and Asn176). While the results of this manuscript describe how molecular recognition paths within a singular SWEET transporter protein differentiate between different substrates, the impact of evolutionary constraints on functional diversification between proteins belonging to different phylogenetic clades cannot be ignored. In the case of SWEETs, the VSVS/ LNMN motif residue positions act as selectivity filters to distinguish functional differences between transporters in clades with alternate preferences for sugar transport. In line with our contact analyses, we evaluate the differences in vdW interaction energy between each sugar and the residues comprising the VSVS motif and nearby binding pocket (Fig. 6). Interaction energies were compared between GLC and SUC bound to the Trp58–Trp180 binding site were compared before (i.e., HG-IF state) and after (i.e., HG-OF state) conformational change for alternate access was

committed. In corroboration with the observation of C–H⋯π stacking (Supplementary Figs. 34 and 35), Trp58 and Trp180 appear to offer stabilization roles for GLC and SUC transport. These stable Trp interactions act as a landing pad to accommodate a variety of C–H⋯π stacking interactions, which may help the sugar substrates sample the necessary facial conformations needed for transport. vdW energies for the rest of the pocket are lower for SUC during the HG-IF state than GLC, maintaining a difference of ~0.25 to 0.84 kcal mol$^{-1}$. After alternate access has occurred, the same trend can be seen where overall vdW interaction energies are again more stable for SUC than GLC.

The greatest discrepancy is for substrate interactions with the VSVS motif. Upon transitioning to an HG-OF state, the SUC interactions with the VSVS motif become more stable. The most stable SUC vdW interactions are those with Val23 (-1.87 ± 0.50 kcal mol$^{-1}$) and Val145 ($-2.74 ± 0.88$ kcal mol$^{-1}$). GLC recognition in the HG-OF state becomes, on average, more stabilized by Val145 ($-1.36 ± 0.98$ kcal mol$^{-1}$) and Val23 ($-0.59 ± 0.52$ kcal mol$^{-1}$). Otherwise, GLC stabilization by Ser54 ($-0.28 ± 0.19$ kcal mol$^{-1}$) and Ser176 ($-0.05 ± 0.03$ kcal mol$^{-1}$) is negligible. Ultimately, our linear interaction energy calculations correspond with experimental observations about the known selectivity preferences of AtSWEET13 for SUC transport. While our simulations depict alternate transport paths and differential levels of sugar–residue interactions, the analyses from this section show how the VSVS motif helps reinforce AtSWEET13 selectivity towards disaccharides.

## Discussion

Just like chemical enzymes, transporters demonstrate facial selectivity. AtSWEET13 was used as a model system to evaluate selectivity principles for membrane transporters. In particular, the SWEET family has piqued much interest with respect to structural and simulation studies[9,18–20,31,46–50]. Our study suggests that chemical selectivity in the AtSWEET13 sugar transporter originates from the protein's ability to optimally position a ligand for recognition before committing to intermediate transitions and alternate access. So long as a ligand's functional group(s) can resemble those of a more native or ancestral substrate, and the protein can recognize it similarly, the ligand can bind, initiate conformational transitions, and be transported. Differences in size do influence the ability of different substrates to sample different facial orientations throughout the transport cycle, where facial selectivity appears to be critical for transport catalysis.

Increases in free energy barriers for *holo* AtSWEET13 sugar transport are surprising, but placing the transport process into a biological context offers clarity. SWEET transporters are bidirectional, cofactor-independent uniporters that catalyze sugar translocation in agreement with a concentration gradient. Catalytic efficiency from reduced effective barriers between intermediate states in response to concentration gradients would also be coupled to an endergonic, downhill, and immediate substrate uptake into cells when required. Critical sink destinations like meristematic cells for embryo development; epidermal cells during stem elongation for cell wall deposition and sugar distribution; stigma and ovary cells for aiding nectar secretion; and trichome cells for the development of defensive sugar-based metabolites would benefit from a seemingly on-demand regulation of sugar allocation[51].

AtSWEET13 alternate access accommodates substrate transport by assuming an HG state where the gating aperture is large enough to house sugar molecules and permit necessary atomistic rotations. Matching its experimentally characterized function, AtSWEET13 transport of GLC is predicted to be a higher energy process than SUC transport. Per the RMSD and tICA decomposition analyses, monosaccharide transport is highly dependent on both protein conformational change and substrate positioning. Meanwhile, disaccharide transport mirrors *apo* transport efficiency in that its transport cycle is highly correlated along a single dimension rather than two (Supplementary Fig. 1). While the extent of protein conformational change between GLC and SUC transport follows similar trends, mono- versus di-saccharide transport pathways bifurcate along early molecular recognition events during IF association. These differences in

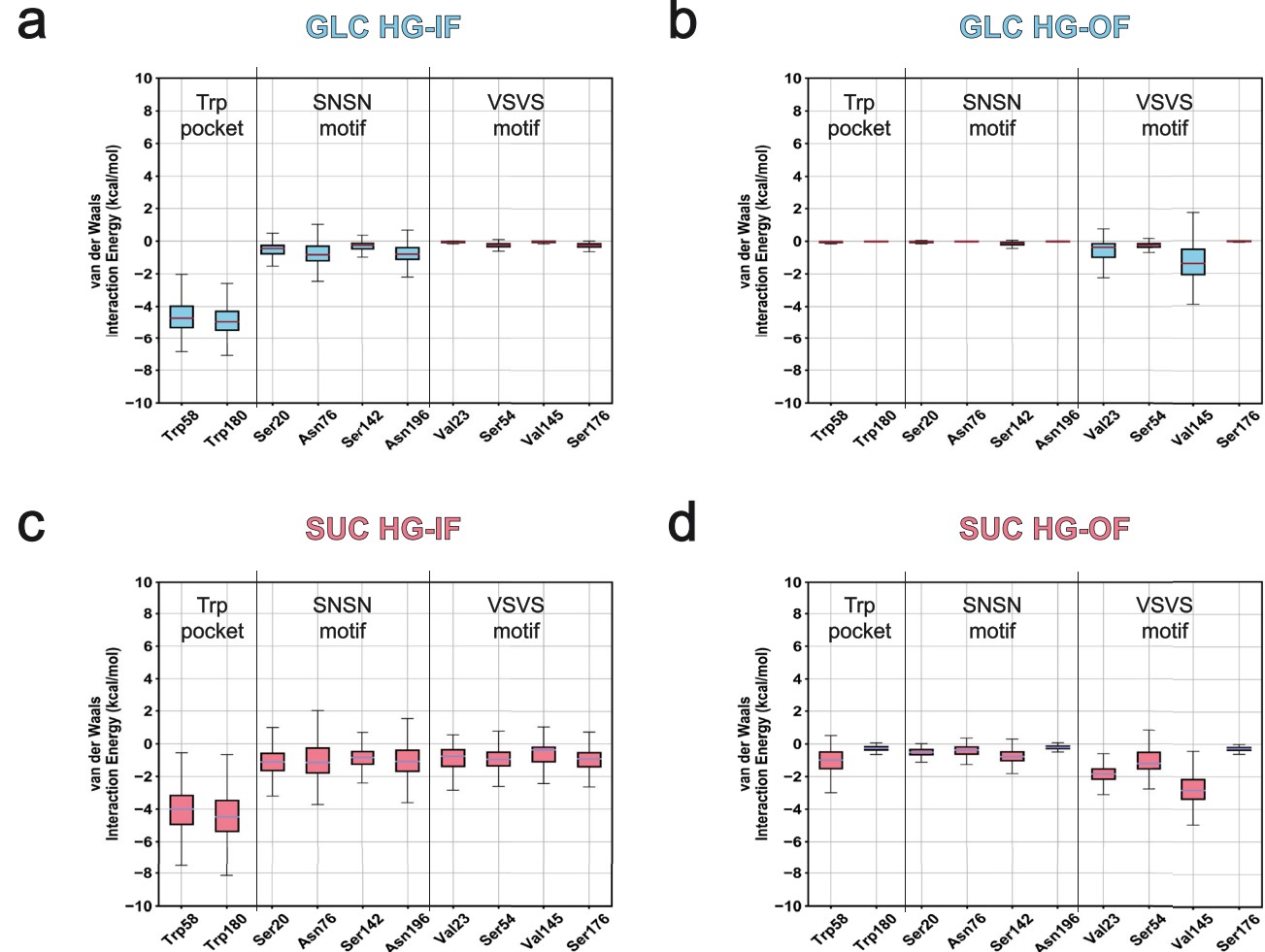

**Fig. 6 | VSVS motif linear interaction energy (LIE) profile comparing GLC and SUC before and after alternate access events.** LIE calculations for interactions between binding pocket residues and GLC before and after alternate access from an HG-IF (**a**) to an HG-OF (**b**) state, respectively. LIE calculations for interactions between binding pocket residues and SUC before and after alternate access from an HG-IF (**c**) to an HG-OF (**d**) state, respectively. LIE error bars are reported as the average ± SD with outliers removed. Number of samples (independent simulation frames) per distribution is reported in the Source Data on our Dryad repository.

functional behavior across the different stages of AtSWEET13 transport were characterized thanks to the label-free atomistic resolution of MD simulations validated by feature-agnostic MSMs.

The ability of AtSWEET13 to invert its substrates in the HG state is likely a reflection of the inverted symmetry gained through the evolutionary advent of TM4 as SWEETs evolved from SemiSWEETs[8,9]. With reference to the topological asymmetry of AtSWEET13, it makes sense that a difference between mono- versus di-saccharide transport is the variation in IF association pathways, as a majority of metastable transport events occur while AtSWEET13 is in an IF-like conformation (Figs. 2 and 4b). Transport catalysis is thus expected to ensue with high efficiency once an intermediate conformational state is achieved with the substrate in an optimal facial orientation. Given the barriers between intermediate and OF states are relatively flat, it can be proposed that the AtSWEET13 substrate transport mechanism mimics the catalysis-promoting strategy of effective intermediate barrier reduction as seen in enzymes utilizing ground state stabilization for rate enhancement[52].

However, differences in translocation pathways for a single protein should not be confused with sequence or structural hallmarks associated with functional specialization seen within a protein family. We have highlighted the specific interactions that are involved during GLC, as well as SUC, recognition and transport. Differences exist with respect to contact frequency, as well as the nature of sugar–residue interactions. Specifically, SUC is seen to enjoy more C–H···π stacking interactions prior to binding in the Trp58–Trp180 binding pocket. Upon reaching the pocket, SUC further enjoys greater stabilization by the disaccharide-preferring VSVS motif as it proceeds through alternate access. Our study offers insights into how cofactor-independent transporters are predicted to accommodate different substrates within the same general transport scheme. It may be possible that the realization of different substrate transport pathways could be leveraged for protein engineering or small-molecule design efforts.

Adding to the difficulty of study based on their non-soluble nature[53], efforts to reveal membrane transporter selectivity principles are complicated by the absence of local catalytic hallmarks as seen in chemical enzymes. Outside of the canonical approach of identifying transporter global conformational changes, we introduce the importance of substrate orientation during the promotion of translocation catalysis by these "physical enzymes" using MD simulation. Ultimately, any molecular design goals surrounding transporters should be dedicated to altering the ability by which critical facial and special intermediate substrate orientations become accessible throughout the transport reaction. In the case of SWEET sugar transport, we hope our efforts help contribute towards the development of blight-resistant crops and improve international food security.

## Methods

### Membrane and membrane protein system assembly

For AtSWEET13 simulations, an inward-facing (IF) *Arabidopsis thaliana* AtSWEET13 crystal structure (PDB: 5XPD) was acquired from the RCSB

Protein Data Bank (PDB)[18,54]. Using CHARMM-GUI, AtSWEET13 was modeled off residues 1–222, where the thermostable mutations performed to aid in crystallization were reverted[18,55–57]. The protein was inserted into a realistic, asymmetric model plant plasma membrane following a maximum complexity compositional recipe (Supplementary Table 7)[48]. Addition of all water and ligand molecules for each system was performed using PACK-MOL 18.169[58]. A 646,400 Å$^3$ volume rectangular box (80 Å$_x$ × 80 Å$_y$ × 101 Å$_z$) was constructed that housed the membrane-embedded AtSWEET13 protein, and 13,439 water molecules were packed into a box. All AtSWEET13 systems were neutralized using KCl to match published experimental conditions[59]. Sugar concentrations were modeled as 100 mM to approximate the sugar concentration of phloem cells as it is reported in plant physiology textbook sources[60,61]. Because of the established sugar concentration, 24 individual sugar molecules were packed into the box.

## System parameterization
All ligands were prepared as PDB files in PyMol, where charge states were determined from p$K_a$ calculations using the ChemAxon Chemicalize tool[62,63]. All ligands were then parameterized by applying the CGenFF36 forcefield using the CHARMM-GUI Ligand Reader & Modeler[64,65]. CHARMM36 protein and lipid forcefields were applied with the psfgen VMD-plugin[66], while the PDB2PQR (PROPKA) server was used to determine protein protonation states[67]. The pH was determined to be 5.3 to replicate the apoplastic side of the plasma membrane in a plant cell. The CHARMM TIPS3P water model was employed, and potassium and chloride ions were used to neutralize the systems using the VMD autoionize package. All systems prepared through VMD were then converted to AMBER format using the ParmEd CHAMBER package (ParmEd 2.7.3)[68]. Hydrogen mass repartitioning was applied to improve statistics by extending single trajectory lengths given available computing wall-clock times[69,70]. VMD 1.9.3 was used in all cases.

## Simulation details
Using the AMBER18 engine[71], the systems were minimized by 5000 cycles of steepest descent and then 45,000 cycles of conjugate gradient with protein backbone atoms restrained. NVT heating from 0 to 10 K was performed over 2 ns, followed by 2 ns of NPT heating at 10 K. Still in NPT, the systems were heated to 300 K over 2 ns, where the temperature was then held at 300 K for 5 ns. All protein residues were restrained during the heating steps. Positional restraints used for heating and minimization were 5 kcal mol$^{-1}$ Å$^{-2}$ in strength. A 50 ns equilibration was then performed without positional restraints. A Langevin thermostat and Monte Carlo barostat were implemented for temperature and pressure maintenance, where the Langevin collision frequency was 2 ps$^{-1}$[72,73]. AtSWEET13 production runs ran for 80 ns. The SHAKE algorithm was applied to all stages of simulation initialization except for minimization[74], while the Particle Mesh Ewald method was used for treating long-range electrostatics at a 12 Å cutoff[75].

## Adaptive sampling protocols
The hallmark of parallelizable simulation seen in adaptive sampling is the ability to select any given state as a starting point for seeding a new trajectory[76–81]. Starting states are selected through a $k$-means clustering of available data given a reaction coordinate of interest, where states occupying the least populated clusters (i.e. least counts clustering) are used for seeding the next round of sampling. The "aggressiveness" of adaptive sampling regimes can be tuned depending on whether clustering and seed selection occur using the data only collected from a given round or across all recorded rounds, as well as the number of seeds generated from each round. For *apo* simulations, distances between gating residues were used as the reaction coordinates for clustering. Monosaccharide simulations were also performed entirely using distances between gating residues. Adaptive sampling metrics for sucrose (SUC) transport changed depending on the stage of transport progress seen in the simulation. At first, distances between gating residues were solely used to promote ligand binding. Once a SUC molecule

was bound and no transport progress was noticeable along the free energy landscape projections after ~3 successive rounds of adaptive sampling, the sampling metric was changed. Atom $Z$ position of the closest SUC molecule to binding site residues Trp58 and Trp180 was then used as the adaptive sampling metric until AtSWEET13 entered an OC/HG-like state. Once SUC was bound to Trp58 and Trp180, the sampling metric was again changed to account for both atom $Z$ position and gating distances. AtSWEET13, extracellular and intracellular gating distances were calculated using residue pairings Lys65–Asp189 and Phe43–Phe164, respectively. From clustering of the provided metrics within the present round, a minimum of 200 states were seeded per round of adaptive sampling for each AtSWEET13 system, where restart coordinate files were prepared using CPPTRAJ (v18.00)[82]. Seeded trajectories were simulated for at least 80 ns in length, depending on available computational resources. It has been shown that using variable lengths of simulated trajectories when constructing a Markov state model does not compromise the quality of the resulting model so long as trajectories are only used, which are at least as long as the working MSM lag time[83]. Starting from the AtSWEET13 crystal structure IF state, adaptive sampling proceeded until OF states were observed, substrate dissociation occurred, and the raw free energy landscapes were well connected (i.e. observable variation in the raw free energy landscape ceases). The use of MDTraj 1.9.3 for measuring distances, VMD 1.9.3. for visualizing states, and Matplotlib 3.2.0 for visualizing projected landscapes were the same as described previously[66,84,85]. In total, 1026 trajectories were generated to resolve the *apo* AtSWEET13 transport cycle; 1465 trajectories were generated to resolve the *holo* GLC AtSWEET13 transport cycle; and 2526 trajectories were generated to resolve the *holo* SUC transport cycle.

## AtSWEET13 atom-specific analyses
Because *holo* sugar simulations were run using multiple copies of the same substrate, the atom under inspection belonging to the ligand molecule found within the AtSWEET13 transmembrane channel was determined. The $Z$ position of the specified atom for the substrate copy undergoing transport was then compared to either the difference between the extracellular and intracellular gating distances or to a rotation angle, theta ($\theta$). For the $\theta$ calculations, imagine a unit circle with a line drawn between points $A$ and $O$, where $A$ represents an atom on the ligand and $O$ represents the graphical origin as the protein ($O_{prot}$) or ligand's ($O_{lig}$) center of mass. The angle between $A$ and $O_{prot}$ in the $XY$ plane ($\theta_{xy}$) represents the specified atom's rotational freedom within the transmembrane channel of AtSWEET13, as the resulting vector signifies directional orientation (and thus contact) between the ligand atom and different regions within the transmembrane channel. Meanwhile, the angle drawn between $A$ and $O_{lig}$ in the $XZ$ or $YZ$ planes ($\theta_{xz}$ or $\theta_{yz}$) indicates the facial orientation of the substrate during different stages of transport (Fig. 4c). Each of these analyses was conducted for every heavy atom (carbon or oxygen) found in each substrate scaffold.

## Determination of agnostic MSM descriptors
Although adaptive sampling regimes typically employ classical MD, the selection of seed states per adaptive round to improve the observation odds of conformational transitions is an inherently biased process. Thus, a fundamental objective for implementing the MSM technique is to statistically validate the aggregate ensemble of MD data through the removal of bias introduced during adaptive sampling seed state selection. A major challenge in the building of MSMs is deciding what features or molecular descriptors will be used for initial data clustering. While there is no exact standard as to how this featurization should be performed, it is generally understood that the MD practitioner should attempt to characterize the slowest dynamic modes while minimizing the use of molecular descriptors directly introduced during adaptive sampling[86–89]. Accordingly, the selection of molecular descriptors for featurization should accurately capture slow dynamical processes relevant to the study objectives while having not been used directly during the adaptive sampling protocol. To this end, representative structures were taken from each relative metastable state along the raw free energy landscapes from select projections. Visual comparison between the

monosaccharide and SUC-free energy landscapes led to the identification of atoms which seemed critical for molecular recognition between the mono- and di-saccharide transport processes. For GLC, this meant collecting states from all C3, C4, O3, and O4 atom-specific landscapes. Critical atoms for SUC state extraction were identified as C2, C5, C6, C9, C10, C11, C12, O5, O6, and O9 landscapes. Metastable energetic minima states were extracted from these landscapes for each substrate dataset, as well as the general extracellular versus intracellular gating landscapes. From all these selected states, C-beta distances between all possible pairwise transmembrane channel residue-residue combinations were calculated (C-alpha was used for glycines). C-beta atoms were selected because of their ability to approximate both backbone and sidechain movement[90]. Using the sci-kit-learn 0.21.2 library, the number of states used to determine the putative MSM features was filtered by identifying which states occupied kmedoids cluster centers. The number of clusters was equivalent to the maximum number of metastable states seen in any individual atom-specific plot for the specified simulation dataset. The inverse of all pairwise channel residue-residue distances was then calculated from this filtered list of metastable states for increased sensitivity[91], and the distances whose standard $z$-score was greater than three standard deviations from the distance distribution mean were selected as initial molecular descriptors for generating a feature-agnostic MSM. With an increased standard $z$-score, it can be assumed that an increased dynamical range (extent of change in the measured distance) has been accurately captured across the extracted metastable states.

### Markov state model (MSM) construction and validation

PyEMMA software (v2.5.6) was used for MSM construction[92]. With exhaustive, agnostic features determined, MSMs were generated using an iterative grid search procedure where the VAMP score was optimized[93]. The MSM protocol used herein can be described as follows:

(1) Given the exhaustive set of agnostic features, construct pilot MSMs to determine the range of cluster counts and time-Independent Components (tICs) capable of returning converged implied timescale plots[94]. From these pilot runs, obtain a working MSM lag time to use for a more exhaustive grid search based on visual inspection of resulting implied timescale plots. A "working" lag time is one that is equivalent to the shortest trajectory timestamp for which a converged implied timescale was observed. By selecting the shortest possible Markovian lag time in this manner, the number of observed jumps within individual trajectories is optimized, and the resolution of MSM processes is maximized.

(2) Perform an exhaustive grid search where the accepted hyperparameter combination optimizes the VAMP score. With this initial MSM transition probability matrix determined, project tIC1 versus tIC2 from the tIC analysis (tICA) object output during clustering.

(3) Confirm that the tICA decomposition landscape from Step 2 is well-connected. Having a well-connected landscape can be indicative of reversible sampling, which is necessary to maintain the Markovian assumption of detailed balance. If the resulting tICA decomposition landscape is well-connected, then determine which features are most correlated with tIC1 and tIC2 for the currently optimized MSM. If not, revisit adaptive sampling to connect this complex tICA landscape with more data.

(4) Once the tICA landscape has been connected, plot the raw cluster counts versus the MSM-weighted cluster counts for the tICA-converged dataset. This validation test ensures that the choice of molecular descriptors used to make the MSM preserves the underlying phase space. Otherwise, resulting free energies may appear exaggerated and result in the misrepresentation of metastable states. If the extent of reweighting between the raw and MSM-adjusted data deviates beyond a relationship of $x = y$ where $y$ is much greater than $x$, then the molecular descriptors used for MSM featurization are deriving an overtuned bias correction (Supplementary Fig. 36)[95,96]. In this case, molecular descriptors that are highly correlated to tIC1 and tIC2 of the

exhaustive tICA decomposition can be used to simplify the MSM featurization step.

(5) Repeat Steps 1–3 using molecular descriptor feature sets inspired by Step 4 until the raw cluster counts versus MSM-reweighting plots approximately satisfy a relationship of $x = y$.

(6) Perform the Chapman–Kolmogorov (C–K) test to ensure that Markovian assumptions of detailed balance are satisfied when constructing an MSM with the working lag time (Supplementary Figs. 37–40).

The final feature sets for each system are listed in Supplementary Table 1. MSM hyperparameters used for each system are described as follows (see Supplementary Table 8). *Apo* MSM hyperparameters: 950 clusters, 3 tICs, a tICA lag time of 8 ns, and an MSM lag time of 30 ns. GLC MSM hyperparameters: 750 clusters, 12 tICs, a tICA lag time of 8 ns, and an MSM lag time of 20 ns. SUC MSM hyperparameters: 900 clusters, 8 tICs, a tICA lag time of 8 ns, and an MSM lag time of 40 ns.

### Adaptive sampling error analysis

The clustering object obtained during the MSM generation protocol was used to obtain 200 data subsets, each containing a random total of 80% of all trajectories contained within each aggregate dataset. From each of these subsets an MSM was prepared. The dimensionality of each data subset was reproportioned to the corresponding number of bins used on the reported MSM-weight free energy landscapes contained within the Main Text of this manuscript. Binning dimensions were selected for each analysis as to maintain a consistent coarse-grained resolution and visual appearance across different analysis types. The standard deviation in reported free energy differences for each of the binned data subsets was used to generate a free energy landscape for sampling error. Bootstrapped free energy error landscapes are provided for all analyses contained within main text figures in Supplementary Figs. 41–43.

### Protein structure visualization and related structural analyses

Protein snapshots were visualized and rendered in Chimera 1.14[97]. RMSD calculations shown in Fig. 2 were calculated using CPPTRAJ and inserted into representative PDB structure $b$-factor columns for worms representation using MDAnalysis 2.0.0[98,99]. Structures selected for the Main Text and Supplementary figures and analyses were chosen from the centers of different metastable minima along relevant free energy landscapes. Selected PDB structures used to calculate transmembrane channel pore radius were analyzed using the HOLE program (v2.2.005)[100]. Contact analyses were performed using GetContacts software[101]. Linear interaction energy analyses were performed using CPPTRAJ commands.

### Statistics and reproducibility

Free energy landscapes figures contain all simulation data reweighted by the resulting MSM weights object. RMSD, LIE, GetContacts, and HOLE analyses are performed on frames obtained from the centers of the metastable states numbered in Fig. 2. Having selected snapshots nearest to the center of each free energy minima, these snapshots will represent some of the most stable states within the free energy landscape surface. Because of this data selection strategy and the statistical reweighting performed by MSMs, the number of frames present within each minima or minima center varies. Candidate frames were extracted from the landscapes by using a coordinate search. Representative frames were taken from the middle of the selection of extracted frames. Representative frames vary in the total count based on the final MSM-reweighting.

Free energy landscapes from simulations can be reproduced by reseeding simulations from each of the provided coordinate files in the Source Data. Free energy landscapes use all obtained simulation data, reweighted by the MSM object.

$t$-test analyses in Fig. 3 were performed using the RMSD distribution averages and standard deviations obtained from the independent MSM-weighted state-representative frames. RMSD analyses were performed by

comparing every individual representative frame from one metastable state as a reference against those from the metastable state corresponding to the preceding stage in transport. For the first stage, State 1, the reference state was the crystal structure 5XPD pose. The average and standard deviation were reported for the RMSD analyses in Fig. 2. Sample sizes varied depending on the number of MSM-reweighted states within each minima and are further detailed in the Dryad deposited Source Data. LIE analyses were performed on frames extracted from HG minima based on landscapes shown in Fig. 2.

GetContacts results were trimmed to report residues that engaged in contact with sugar substrates for at least 1% of all representative frames being analyzed.

## Reporting summary

Further information on research design is available in the Nature Portfolio Reporting Summary linked to this article.

## Data availability

The data for this study, including the source data used to generate the graphs presented in Fig. 6, has been deposited to a Dryad repository [https://doi.org/10.5061/dryad.8sf7m0cxn]. The repository contains simulation and simulation-related results and python scripts; jupyter notebooks; MSM objects and PDBs of select AtSWEET13 conformational states. All such files are submitted as Source Data to the listed Dryad repository. The crystal structure of AtSWEET13, PDB accession code 5XPD, was referenced in the Main Text of this document [5XPD]. Simulation and simulation-related results and python scripts; jupyter notebooks; MSM objects and PDBs of select AtSWEET13 conformational states are available at the Shukla Group github at the following web address: https://github.com/ShuklaGroup/Weigle_Shukla_Communications_Biology_2024/.

## Code availability

Codes and files used in this manuscript have been made available. Codes used in this manuscript are provided in a Box link, which is included in the project description for the following github page. The github is version controlled through our Dryad repository [https://doi.org/10.5061/dryad.8sf7m0cxn]. The link is embedded with public access privileges, meaning that clicking on the provided Box link will reveal access to the codes. The Box link can be found from our github page: https://github.com/ShuklaGroup/Weigle_Shukla_Communications_Biology_2024/.

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

## Acknowledgements

This research was part of the Blue Waters sustained-petascale computing project, supported by the National Science Foundation (awards OCI-0725070 and ACI-1238993), the State of Illinois, as of December 2019, the National Geospatial-Intelligence Agency. During its tenure, Blue Waters was a joint effort of the University of Illinois and its National Center for Super-computing Allocations. We acknowledge support from the National Institutes of Health (Award No. R35GM142745). We thank Soumajit Dutta for useful discussions concerning MSM validation and Arnav Paul for suggesting additional literature sources for citation.

## Author contributions

D.S. acquired funding for the project. D.S. conceived and supervised the project. A.T.W. performed modeling and simulations. A.T.W. and D.S. analyzed the simulation data. A.T.W. wrote the manuscript with inputs from D.S., where both authors have expressed their approval of the final version of this manuscript.

## Competing interests

The authors declare no competing interests.
