## [Peer Review File · Communications Biology]

Reviewers' comments:

Reviewer #1 (Remarks to the Author):

The authors comprehensively describe the results of a series of molecular dynamics simulations focused on explaining the mechanism of glucose versus sucrose molecular recognition and transport mediated by AtSWEET13 (transmembrane sugar transported protein). The manuscript is well-conceived and well-written, and I commend the authors on the thoroughness of their work. Indeed, this is one of the best molecular modeling-oriented manuscripts I have reviewed in the past several years. The manuscript may benefit from minor corrections, according to suggestions given below. Aside from that, I do not have any criticisms.

* The authors describe in detail the mechanism of protein conformational changes associated with sugar transport but very little is said about the nature of interaction at the sugar/protein interface. What is the driving force for binding sugar molecules at their pathways within AtSWEET13? For instance, it is shown that some tryptophans are responsible for formation of binding pocket and sugar binding. Do related contacts results from CH- π interactions? Can some conclusion be drawn about behavior of glucose-like molecules, characterized by smaller magnitude of CH- π interactions with Trp? I recommend to briefly discuss the issue of sugar-protein interactions in the context of reported findings.

* I also recommend to explicitly describe the system composition in the Methods section (number of solvent molecules, box size and shape and, most importantly, the composition of lipid membrane).

* The letter code for amino-acid residues should be unified.

Reviewer #2 (Remarks to the Author):

Weigle and Shukla have performed extensive molecular dynamics simulations with enhanced sampling to explain how the Arabidopsis sugar transporter AtSWEET13 differentiates between mono- and disaccharides. Even though they have collected a significant amount of data through these simulations, the manuscript fails to deliver its promise. My major concerns are: (1) the current simulations and analysis fail to provide a comprehensive understanding of glucose versus sucrose recognition and transport, (2) the data and the results are very poorly presented.

A. As reported in their Methods section, Weigle and Shukla use the AtSWEET13 crystal structure and revert the thermostable mutations performed to aid in crystallization and run simulations with the wild type AtSWEET13 (WT-AtSWEET13) (p. 20). However, as revealed in the original article reporting this crystal structure (Han et al. (2017) PNAS), a very important point is that these residues are known to determine mono- or disaccharide specificity. V23, S54, V145 and S176 ("VSVS", WT-form) are commonly found in disaccharide transporters, whereas L23, N54, M145 and N176 ("LNMN") are found in monosaccharide transporters. The WT is already known to be selective for disaccharides, and these

residues in the WT ("VSVS") dictate disaccharide specificity. Specific concerns regarding the conclusions:

1. One of the main findings of the paper (Results sections 1 and 2) is that binding events are more stable for sucrose than for glucose, and that transport of glucose is a higher energy process than sucrose transport. As they have used the WT-AtSWEET13 known to be selective for disaccharides, this is an expected result. It is great that the simulations agree with this, but it is not a novel finding.
2. Results section 3 is the section that claims to explain discriminative sugar recognition. What is the main finding of this section? How does AtSWEET discriminate between these sugars? The authors provide atom-specific analysis, but what is the outcome? This section has to be completely revised as the current atom-specific results are not of relevance to the experts in this field, let alone for a broader audience that this journal targets.
3. There are no results/data presented in the Results section 4 titled "Exploiting evolutionary context for mechanistic generalization and predictive engineering". The authors mention that "to make AtSWEET13 more selective for either GLC or SUC would then imply the need for mutations which would lower the effective barrier between the IF ground state and the flat intermediary free energy landscape." Can they show the experimental validation for this where they perform mutagenesis experiments to make the transporter more selective for glucose or sucrose (by reporting glucose/sucrose transport with these mutations). If they cannot add any experimental results to this section, it should be removed from the Results section.
4. Related to this point, I find the Discussion section very confusing and the conclusions handwaving. The authors claim that their study offers insights into "how general transport selectivity can be engineered." (p. 18) If they cannot support their claim in the Results section 4, they should remove this sentence as well.
5. The authors claim that their proposed mechanism "suggests AtSWEET13 can tap into some 'primordial memory' to first position substrates based on IF molecular recognition of glucose-like atoms" (p. 15). Can they show that their simulations can in fact recognize sugar molecules known not to bind this transporter? If they cannot support this claim, they should remove it.

B. Additional questions about the design of the study:

1. As mentioned in the above section, there are two sets of variants of the four residues that are known to dictate selectivity. Can the authors run additional sets of simulations to compare the results (i) with VSVS, (ii) with LNMN? This comparative study might lead to findings that are more generalizable to the AtSWEET family.
2. The authors mention that in addition to glucose and sucrose, giberellin is also a substrate of AtSWEET13. Can the authors explain why they chose to focus on glucose and sucrose, and not giberellin?
3. Han et al. have shown a dimer unit transport model. Can the authors comment on why they have considered to focus on the monomer model instead the dimer model?

C. The authors should organize their data and find better ways to present and summarize their data in a concise and elegant fashion. There are 40 supplementary figures with multiple subpanels. Their supplementary file looks like a "data dump". Similar to the figures, the text is also very lengthy and needs major improvement. Specific questions regarding the presentation of the data:

1. What is the point of including bar plots for the tIC-correlating features (Figs. S15-S17) where all the bars show high values? Is the correlation value for each bar significant to the story? What is the intended message of all of these bar plots?

2. The authors have projected RMSD values onto the protein structure in Figs. 2 and S9. It is very difficult to understand the significance of these images as the colors of the most important regions are very close to one another. I was not able to confirm the claim that "the average per-residue RMSD is greater for overall GLC transport than for SUC" (p. 6), as the RMSD of terminal residues are the highlight of these figures, which is not of interest. Instead of projecting the values on the structures, the authors should use RMSD plots to evaluate whether there is a significant difference between glucose and sucrose. They should specify which regions show the greatest difference and most importantly, assess whether this difference is significant.

3. In Figs. 2A and 2B, the authors show plots linking the difference in intracellular versus extracellular gating to the sugar position. They also plot this for other atoms of the sugar molecules in Figs. S3-S8. What is the reasoning behind choosing glucose O5 and sucrose O3 atoms to represent all of these plots in the main figure? The only reason given by the authors was that "they represent the same ether oxygen along nearly identical hexose units" (p.10). Considering that there are other oxygens in identical positions in glucose and sucrose, why was this oxygen chosen? Similarly, what is the reasoning behind choosing the structures in Fig. 2C and 2D among all the other ones in Fig. S9? Similarly, what is the reasoning behind choosing the panels in Fig. 3B among all the other ones in Fig. S18-S35?

4. Following up on point 3, instead of producing many plots that essentially show the same thing (for example the plots they produced using the z-coordinates of each sugar atom in Figs. S3-S8) the authors should use more comprehensive metrics that can illustrate their finding (e.g. using the z-coordinate for the center of mass of glucose).

5. Most of the data is presented in heat maps (Figs. 1B, 2A, 2B, 3B, S1-S8, S18-S35, S37-S39). The heat maps in the main figures contain numbers and arrows, but it is not clear what these numbers mean or which states the numbers correspond to. The heat maps in the supplementary do not contain any labels and it is unclear what the reader should take away from all of these plots. Such plots should be accompanied by representative conformational snapshots. Again, better metrics should be defined to present the results in a concise way and different modes of data visualization techniques should be used to facilitate understanding (e.g. chord diagrams for free energy where each state represents a node, with conformational snapshots linked to each node).

6. Fig. 4 is very difficult to understand as the transporter is shown from different angles in the snapshots. The authors should improve this to show the transporter from the same side view angle in each snapshot and use slab representations to show the different states and ligand association/dissociation. If needed, they should use zoomed-in insets to show the ligand orientation.

7. The second section of Results is titled "Conformational change cannot explain differences in AtSWEET13 sugar transport". The authors should not focus on the aspects that cannot explain the differences in sugar transport, but instead concisely explain which aspects can.

8. The text needs to be revised to remove the sections about the details of the methodology. Only brief descriptions of the methodology should be included in the Results section, and these descriptions should be understandable by the broader audience of the journal.

D. The rigor of the simulations should be improved as well. Specific concerns regarding the rigor of simulations:

1. At what pH were the protonation states determined? What was the protonation state of D189? How were the histidines modelled (protonated or neutral, and if neutral, in which tautomeric state (δ/ϵ)?)

2. Figs. 1, 2, and 4 include conformational snapshots from simulations. How were these conformational snapshots selected? Can the authors show tests to ensure that these are representative?
3. Which form of glucose was modelled (D/L-glucose and alpha/beta-glucose)? Was this the most relevant form for this transporter?
4. The authors stated that "All holo AtSWEET13 systems were built using 100 mM of substrate to match published experimental conditions." (p. 20). Can the authors point to the location where in the referenced article this is mentioned? Can the authors mention how many sugar molecules were in the simulation box?
5. How were the lag times determined? Which tests were performed for model validation?
6. The authors have mentioned "seeded trajectories were simulated for at least 80 ns in length, depending on available computational resources." (p.22) How long were the simulations?

E. Minor comments:

1. In Fig. 1, the authors show the apo AtSWEET13 transport cycle. It would be more compelling if the authors instead combine the holo and apo transport cycles showing ligand association/dissociation steps. In essence, Fig. 1A and Fig. 4 show the same cycle and it is confusing why the cycle is shown multiple times.
2. There are many instances in the text where the authors have referred to the wrong supplementary figure, which should be fixed.
3. By convention, the chair form of a sugar is drawn with the ring oxygen at the back right position. The authors should use the conventional chair notation in their figures, or explain their reasoning if they choose not to.

Reviewer #1 (Remarks to the Author):

The authors comprehensively describe the results of a series of molecular dynamics simulations focused on explaining the mechanism of glucose versus sucrose molecular recognition and transport mediated by AtSWEET13 (transmembrane sugar transported protein). The manuscript is well-conceived and well-written, and I commend the authors on the thoroughness of their work. Indeed, this is one of the best molecular modeling-oriented manuscripts I have reviewed in the past several years. The manuscript may benefit from minor corrections, according to suggestions given below. Aside from that, I do not have any criticisms.

We greatly thank and appreciate the comments by Reviewer 1 for seeing purpose in our work. In these revisions, we provide additional details on AtSWEET13-sugar interactions, per the request of Reviewer #1.

- 1. The authors describe in detail the mechanism of protein conformational changes associated with sugar transport but very little is said about the nature of interaction at the sugar/protein interface. What is the driving force for binding sugar molecules at their pathways within AtSWEET13? For instance, it is shown that some tryptophans are responsible for formation of binding pocket and sugar binding. Do related contacts result from CH- π interactions? Can some conclusion be drawn about behavior of glucose-like molecules, characterized by smaller magnitude CH- π interactions with Trp? I recommend to briefly discuss the issue of sugar-protein interactions in the context of reported findings.**

We thank Reviewer #1 for bringing up how the work needed a greater structural biology focus, and that the binding interactions specific to each sugar needed further discussion.

In response, we added a section called “*Describing distinct binding pathways for glucose versus sucrose sugar transport pathways*”. We added a GetContacts analysis to see which residues participated in sugar molecular interactions during different stages of transport. For the most part, we identified van der Waals contacts and C-H $\cdots\pi$ stacking interactions. The GetContacts analyses are summarized as heatmaps in Supplementary Figures 32 and 33. We also provide snapshots for representative instances of C-H $\cdots\pi$ stacking interactions during carbohydrate substrate interaction. These snapshots are provided in Supplementary Figures 34 and 35.

Here we provide the new Supplementary Figures and their captions:

Supplementary Figure 32. GetContacts analysis for representative frames of metastable states throughout GLC transport. Metastable states are the same as those enumerated from Main Text Figure 4b.

Supplementary Figure 33. GetContacts analysis for representative frames of metastable states throughout SUC transport. Metastable states are the same as those enumerated from Main Text Figure 4b.

Supplementary Figure 34. C-H $\cdots\pi$ stacking and other interactions between GLC and AtSWEET13 throughout molecular recognition and transport. **(a)** GLC binding during early IF recognition along intracellular gate residue Phe43. **(b)** GLC-Phe43 interactions facilitate further substrate entry into the transmembrane channel. **(c)** Preparative GLC binding interactions during an extended HG-IF-like state before commitment to alternate access. **(d)** Extensive GLC C-H $\cdots\pi$ stacking with the Trp58-Trp180 binding pocket. Goldenrod spheres are used to represent the centers of mass of aromatic systems within residue side chains. C-H $\cdots\pi$ stacking is indicated using vertical black bars between substrate hydrogen atoms and the aromatic goldenrod spheres.

Supplementary Figure 35. C-H $\cdots\pi$ stacking and other interactions between SUC and AtSWEET13 residues throughout molecular recognition and transport. **(a)** SUC binding during early IF recognition along intracellular gate residues Phe43 and Phe164. **(b)** Further SUC entry into the intracellular-facing vestibule of the transmembrane channel. Preparative SUC binding interactions during an extended HG-IF-like are stabilized by C-H $\cdots\pi$ stacking with either **(c)** Phe24 or **(d)** Phe146. **(e)** SUC-Phe43 C-H $\cdots\pi$ stacking maintains an HG conformation during required facial inversion. **(f)** SUC enjoys extensive C-H $\cdots\pi$ stacking throughout commitment to alternate access. Goldenrod spheres are used to represent the centers of mass of aromatic systems within residue side chains. C-H $\cdots\pi$ stacking is indicated using vertical black bars between substrate hydrogen atoms and the aromatic goldenrod spheres.

We added a few citations from the lab of Prof. Jesús Jiménez-Barbero to help frame discussion around the relative role of sugar-aromatic stacking forces in the AtSWEET13 molecular recognition process. These citations include:

[43] Solís, D., Romero, A., Menéndez, M. & Jiménez-Barbero, J. Protein-carbohydrate interactions: Basic concepts and methods for analysis. in *The Sugar Code* (ed. Gabius, H.-J.) 223–245 (WILEY-VCH VerlagGmbH & Co. KGaA, 2009).

[44] Ardá, A. & Jiménez-Barbero, J. The recognition of glycans by protein receptors. Insights from NMR spectroscopy. *Chem. Comm.* **54**, 4761–4769 (2018).

[45] Asensio, J. L., Ardá, A., Cañada, F. J. & Jiménez-Barbero, J. Carbohydrate–Aromatic Interactions. *Acc. Chem. Res.* **46**, 946–954 (2013).

2. I also recommend to explicitly describe the system composition in the Methods section (number of solvent molecules, box size and shape and most importantly, the composition of lipid membrane).

Additional information has been added about the system size, the number of solvent molecules, the box volume/size/shape, and the composition of the lipid membrane. This information has been added to the Methods subsection “*Membrane and membrane protein system assembly*”:

“For AtSWEET13 simulations, an inward-facing (IF) *Arabidopsis thaliana* AtSWEET13 crystal structure (PDB: 5XPD) was acquired from the RCSB Protein Data Bank (PDB).^{18,54} Using CHARMM-GUI, AtSWEET13 was modeled off residues 1-222, where the thermostable mutations performed to aid in crystallization were reverted.^{18,55–57} The protein was inserted into a realistic, asymmetric model plant plasma membrane following a Maximum Complexity compositional recipe (Supplementary Table 8).⁴⁸ Addition of all water and ligand molecules for each system was performed using PACKMOL 18.169.⁵⁸ A 646,400 Å³ volume rectangular box (80 Å_x x 80 Å_y x 101 Å_z) was constructed that housed the membrane-embedded AtSWEET13 protein, and 13439 water molecules were packed into the a box that was. All AtSWEET13 systems were neutralized using KCl to match published experimental conditions.⁵⁹ Sugar concentrations were modeled as 100 mM to approximate the sugar concentration of phloem cells as it is reported in plant physiology textbook sources.^{60,61} Because of the established sugar concentration, 24 individual sugar molecules were packed into the box.”

3. The letter code for amino-acid residues should be unified

The letter code for amino acids has been changed to be three characters, rather than one, throughout all text and figures.

Reviewer #2 (Remarks to the Author):

Weigle and Shukla have performed extensive molecular dynamics simulations with enhanced sampling to explain how the Arabidopsis sugar transport AtSWEET13 differentiates between mono- and disaccharides. Even though they have collected a significant amount of data through these simulations, the manuscript fails to deliver its promise. My major concerns are: (1) the current simulations and analysis fail to provide a comprehensive understanding of glucose versus sucrose recognition and transport, (2) the data and the results are very poorly presented.

We apologize for our previous submission having disappointed Reviewer #2. However, we appreciate the opportunity to improve the presentation and packaging of our work, specifically with a greater emphasis that would be of value to structural biology communities. We have expanded upon the scope of the manuscript given recommendations by Reviewer #2.

A. As reported in their Methods section, Weigle and Shukla use the AtSWEET13 crystal structure and revert the thermostable mutations performed to aid in crystallization and run simulations with the wild type AtSWEET13 (WT-AtSWEET13) (p. 20). However, as revealed in the original article reporting this crystal structure (Han et al. (2017) PNAS), a very important point is that these residues are known to determine mono- or disaccharide specificity. V23, S54, V145, and S176 (“VSVS”, WT-form) are commonly found in disaccharide transporters, whereas already known to be selective for disaccharides, and these residues in the WT (“VSVS”) dictate disaccharide specificity. Specific concerns regarding the conclusions:

Our original submission focused on the differences in the molecular recognition and transport paths for GLC and SUC transport by wild-type AtSWEET13. As our manuscript was submitted, we find our original results to be worthy of dissemination because they describe wild-type AtSWEET13 function. However, we do agree with Reviewer #2’s suggestion to compare against the LNMN AtSWEET13 variant for two reasons:

1. Studying the VSVS versus LNMN forms of AtSWEET13 could enable broader understanding of the SWEET transporter family.
2. The site for the VSVS/LNMN mutations is near the conserved W58-W180 binding site. From our simulation results and atom-specific analyses, we observe that the spatial positioning of sugar moieties is conserved at this point, reflecting the importance of a type of “transition state” during sugar translocation catalysis. The authors from the Han *et al.* (2017) PNAS (<https://www.pnas.org/doi/10.1073/pnas.1709241114>) paper also show that a W58A mutation significantly reduces sucrose transport, while W180A mutation abolishes sucrose transport. From our atom-specific analyses, we can observe how this mutation affects the stabilization of sugars at the transition state to alternate access.

Per the request of Reviewer #2, we have performed an additional set of simulations to observe how the LNMN mutation set would impact the *holo* glucose and sucrose transport cycles of AtSWEET13.

The original submission of this manuscript took ~2 years to complete. In order to make the capture of the AtSWEET13-LNMN mutant transport cycle more efficient, we selected a random state from each of the AtSWEET13-WT microstate clusters in the final optimized Markov state models for AtSWEET13 GLC and SUC transport datasets. We performed these simulations on Folding@Home. Following equilibration of each of the states, we then seeded each state in triplicate for 160ns. We doubled the trajectory length from 80ns to 160ns and performed each seed state simulation in triplicate in the hopes that we would obtain sufficient sampling with just a single project launch for LNMN-GLC and LNMN-SUC simulations.

In total, our datasets included 2246 trajectories of LNMN-GLC (359.4 μ s) and 2698 trajectories of LNMN-SUC (431.68 μ s) transport. Altogether, our additional simulations totaled to an ~791 μ s of added work.

To meet the deadlines for revisions (which we are already late to), we had to approach this comment by means of selecting seed states from the AtSWEET13-WT transport cycles. To our dismay, the selection of seed states from the AtSWEET13-GLC and AtSWEET13-SUC transport pathways introduced a statistical bias into the AtSWEET13-LNMN results. Most of our findings remained the same from our original submission with respect to the wild-type results.

The primary clue towards forming our conclusive distrust of this mutant data was when we reviewed our AtSWEET13-LNMN SUC atom-specific rotation plot analyses. Specifically, our inversion analyses showed that SUC atoms demonstrated free inversion throughout transport. Equivalent access to both inversion pathways is highly unlikely, as within the AtSWEET13-WT SUC atom-specific analyses, access to the alternate inversion pathway incurred an energetic penalty of ~6-7 kcal/mol in MSM-weighted relative free energy. While we expected that the path for AtSWEET13-LNMN SUC transport may change as a result of mutation, we did not expect the WT path to be preserved at an equally lower energy. Statistical oversampling of both these paths resulted in a misrepresentation of the SUC transport, suggesting that AtSWEET13-LNMN mutations would possess a lower barrier for SUC transport than AtSWEET13-WT, which is not the case per the *Han et al. (2017) PNAS* paper. To overcome this limitation in methodology based on the timeline for peer review, we attempted to try and describe the linear interaction energies of the different sugar poses within the binding site for the AtSWEET13-WT and AtSWEET13-LNMN mutations. To our disappointment, the linear interaction energies were nearly identical for all sugar-residue interactions and again reflected the fundamental issue with choosing seed states from the AtSWEET13-WT path for sampling LNMN mutant transport.

While we have attempted to address your request, the simulation of AtSWEET13-LNMN transport was not feasible within the timeline for peer review. However, we do realize that Reviewer #2 felt that our conclusions concerning the discriminatory nature of the different

sugar binding pathways was overreaching. Our omission of the VSVS/LNMN motifs discussed in the *Han et al. (2017) PNAS* paper was not intentional. We now discuss the importance of these motifs, and distinguish this importance apart from our findings related to binding paths.

We approach the importance of the VSVS motif by discussing it in context of the different interactions seen during GLC versus SUC binding. This discussion occurs in the new Results section titled “*Describing distinct binding pathways for glucose versus sucrose sugar transport pathways*”. With respect to the role of the VSVS selectivity filter, linear interaction energy analyses comparing sugar-protein contacts before and after commitment access show that the VSVS residues have more stabilizing interactions with SUC as opposed to GLC. We show these results in our new Figure 6:

Figure 6. VSVS motif linear interaction energy (LIE) profile comparing GLC and SUC before and after alternate access events. LIE calculations for interactions between binding pocket residues and GLC before and after alternate access from an HG-IF (a) to an HG-OF (b) state, respectively. LIE calculations for interactions between binding pocket residues and SUC before and after alternate access from an HG-IF (c) to an HG-OF (d) state, respectively.

We feel that these results are in line with the expectations of Reviewer #2. We also now provide disclaimers to our results by treating our rotational analyses with an importance separate from the critical role of the VSVS/LNMN filter in demarcating selectivity throughout the SWEET family. We write in this new results subsection:

“Our identification of separate intracellular binding pathways highlights how a single SWEET transporter which transports different sugar types may demonstrate selectivity preferences. However, overall selectivity between members within a transporter family are likely to be associated with functionally divergent phylogenetic profiles. In the original work resolving the AtSWEET13 crystal structure, Han *et al.* used multiple sequence alignments to identify that a subgroup of residues near the Trp58-Trp180 binding pocket whose patterns appeared to dictate substrate selectivity between different SWEET transporters.¹⁸ Specifically, AtSWEET13 residues Val23, Ser54, Val145, and Ser176 constitute a “VSVS” motif common to disaccharide transporters. Meanwhile, solely monosaccharide-transporting SWEETs exhibit an “LNMN” motif along the same residues (i.e., Leu23, Asn54, Met145, and Asn176). While the results of this manuscript describe how molecular recognition paths within a singular SWEET transporter protein differentiate between different substrates, the impact of evolutionary constraints on functional diversification between proteins belonging to different phylogenetic clades cannot be ignored. In the case of SWEETs, the VSVS/LNMN motif residue positions act as selectivity filters to distinguish functional differences between transporters in clades with alternate preferences for sugar transport.”

- 1. One of the main findings of the paper (Results sections 1 and 2) is that binding events are more stable for sucrose than for glucose, and that transport of glucose is a higher energy process than sucrose transport. As they have used the WT-AtSWEET13 known to be selective for disaccharides, this is an expected result. It is great that the simulations agree with this, but it is not a novel finding.**

Our intention of including such a sentence was to increase trust in our results, suggesting that they do corroborate with experiments.

While it is known that WT-AtSWEET13 is already known to be selective for disaccharides, it is not understood *why* it is selective. Our original language with respect to glucose being higher energy than sucrose transport is descriptive. It is included to ground our simulation results with experiments. However, this difference in free energy merits an explanation, which we attempt by describing the tICA feature set reduction. These Supplementary Figures (originally Figures S13-S17; see Reviewer #2 *Comment C.1.*) present all highly correlated features and their levels of correlation to each respective transport process. They are attempts to describe some of the features that are behind this difference in free energy.

When viewing Reviewer #1's comments with Reviewer #2's *Comment A.1.* retrospectively, we also recognize that our manuscript needs to include greater description

of protein dynamics and protein-sugar interactions along the different binding pathways. We outline such made changes throughout this Rebuttal Letter.

- 2. Results section 3 is the section that claims to explain discriminative sugar recognition. What is the main finding of this section? How does AtSWEET discriminate between these sugars? The authors provide atom-specific analysis, but what is the outcome? This section has to be completely revised as the current atom-specific results are not of relevance to the experts in this field, let alone for a broader audience that this journal targets.**

Concerning relevance of Results Section 3

We will first respond to Reviewer #2's commentary on the relevance of Results Section 3.

Membrane transporters exist within a unique intersection of interests across the chemical biology spectrum. To name a few interested parties, study of membrane transporters attracts molecular biologists, medicinal chemists, structural biologists, chemical biologist, biophysicists, biochemists, molecular modelers and computational scientists. The peer reviewers chosen by the editorial board for this manuscript should belong to one of these interested parties.

Provided one of the summary comments by Reviewer #1, "*The manuscript is well-conceived and well-written, and I commend the authors on the thoroughness of their work. Indeed, this is one of the best molecular modeling-oriented manuscripts I have reviewed in the past several years.*" Results Section 3 does appear to appeal to researchers within the broader audience that this journal targets. Hence, we respectfully maintain this analysis in our manuscript resubmission. However, we have provided greater analyses which offer more relevance to structural insights related to AtSWEET13 conformational change and molecular recognition in other Results sections.

Concerning the outcome of Results Section 3

We agree with Reviewer #2 that a specific outcome from our atom-specific analyses should be better outlined. Our Results Section 3 is mainly describing how transport differs between GLC and SUC from the perspective of the ligand. As Reviewer #1 also asks, we should include more details about specific binding interactions from the perspective of the protein. In response, we have created a new Results Section "***Describing distinct binding pathways for glucose versus sucrose sugar transport pathways***". We describe this newly introduced section in your *Comment A.3*. In this new section, we describe the binding pathway on the basis of local residue interactions. We also revise the original Figure 4 (*see Comment C.6.*). Although we are now addressing the aspects of residue-sugar interactions in another section, that does not mean that our Results Section 3 lacks a main finding in and of itself, as Reviewer #2 suggests.

The main finding of this section is that despite AtSWEET13 sampling the same conformational ensemble for both GLC and SUC transport, the substrates experience different transport pathways as they proceed from intracellular binding events through alternate access and export. This analysis identifies the structural portions within the AtSWEET13 transmembrane channel where the sugar transport pathways differ. To directly quote our original submission:

“GLC IF association then must proceed within the range of $135^\circ \leq \theta_{xy} \leq 225^\circ$, as the GLC C3 atom samples transmembrane channel space between TMs 5, 6 and 7 ... inspiring us to refer to this rotational space as part of the ‘monosaccharide pathway’.”

“In contrast to GLC, the SUC₂ state therefore leads the GLC-like atom towards the TM6-TM1 cleft, space which the equivalent atom in the GLC structure does not approach during the same stage of transport. Thus, we term this area of AtSWEET13 as part of the “disaccharide pathway.”

Given the differences in the pathways and their relative free energy, we conclude that “AtSWEET13 specialization in disaccharide transport emphasizes the importance of pentose molecular recognition during HG-OF transitions and OF dissociation.”

We agree that after rereading our original manuscript, this section is deserving of a more conclusive paragraph/outro. We provide it here:

“Overall, these atom-specific rotational analyses reveal that despite AtSWEET13 sampling the same conformational ensemble during GLC and SUC transport, the substrates experience different transport pathways. By considering transport from the perspective of the ligand, these analyses enable identification of the structural portions within the AtSWEET13 transmembrane channel where the sugar transport pathways differ. These functional differences are also reflected in the perception of hexose and pentose units within SUC (Supplementary Figures 12-29). Specifically, IF molecular recognition of the SUC hexose unit emphasizes that it can be perceived like monomeric GLC by potentially sampling the monosaccharide path, but that this mode of transport is likely to be less probable based upon MSM-weighted relative free energy calculations. AtSWEET13 specialization in disaccharide transport therefore emphasizes the importance of pentose molecular recognition during HG-OF transitions and OF dissociation.”

3. **There are no results/data presented in the Results section 4 titled “Exploiting evolutionary context for mechanistic generalization and predictive engineering.” The authors mention that “to make AtSWEET13 more selective for either GLC or SUC would then imply the need for mutations which would lower the effective barrier between the IF ground state and the flat intermediary free energy landscape.” Can they show the experimental validation for this where they perform mutagenesis experiments to make the transporter more selective for glucose or sucrose (by reporting glucose/sucrose transport with these mutations). If they**

cannot add any experimental results to this section, it should be removed from the Results section.

For reasons outside of our control we were unable to include experiments in our submission of this manuscript. As such, we agree with Reviewer #2 and have removed specifications towards engineering strategies that should be reserved for future studies supported by experiments.

- 4. Related to this point, I find the Discussion section very confusing and the conclusions handwaving. The authors claim that their study offers insights into “how general transport selectivity can be engineering.” (p. 18) If they cannot support their claim in the Results section 4, they should remove this sentence as well.**

Per the request of Reviewer #2, we removed discussion points related to strategies towards engineering AtSWEET13 and reserve that discussion for future studies.

- 5. The authors claim that their proposed mechanism “suggests AtSWEET13 can tap into some ‘primordial memory’ to first position substrates based on IF molecular recognition of glucose-like atoms” (p. 15). Can they show that their simulations can in fact recognize sugar molecules known not to bind to this transporter? If they cannot support this claim, they should remove it.**

The referenced text has been removed.

B. Additional questions about the design of the study:

- 1. As mentioned in the above section, there are two sets of variants of the four residues that are known to dictate selectivity. Can the authors run additional sets of simulations to compare the results (i) with VSVS, (ii) with LNMN? This comparative study might lead to findings that are more generalizable to the AtSWEET family.**

Please refer to our direct response to the header of *Comment A*.

- 2. The authors mention that in addition to glucose and sucrose, gibberellin is also a substrate of AtSWEET13. Can the authors explain why they chose to focus on glucose and sucrose, and not gibberellin?**

These simulations are expensive. To put things into perspective, the amount of simulation included in this study took 231,408 hours, or approximately 26.42 years, in human time. The key question for this study, as submitted, is understanding the difference between SUC and GLC transport, rather than gibberellin.

When we originally designed this study, we wanted to perform simulations of gibberellin transport with AtSWEET13 as well. We also intended to include experiments. However,

because of the pandemic, timelines for both experimental collaboration and gibberellin transport simulations changed. In response, we wanted to create a simple story where we focused on what could be more “canonical” transport behavior by AtSWEET13. This made the paper focus dedicated to sugar transport.

Plants do not have a dedicated gibberellin transporter. That is, all known gibberellin transporters are non-specific, as their primarily identified function is designated for transport of some other substrate {see Dayan, J. Gibberellin Transport. in *The Gibberellins, Volume 49*, (eds. Hedden, P. & Thomas, S. G.) pages (Annual Plant Reviews, 2018). <https://doi.org/10.1002/9781119312994.apr0533>}. It is hypothesized that transport of gibberellin is predicated off biological need (i.e., the developmental state and needs of a plant at that time “require” certain transporters to transport gibberellins) {see Gantait, S., Sinniah, U. R., Ali, M. N. & Sahu, N. C. Gibberellins – a multifaceted hormone in plant growth regulatory network. *Curr. Protein Pept. Sci.* **16**, 406-412. (2015) DOI: <https://doi.org/10.2174/1389203716666150330125439>}. We are currently in progress of preparing this gibberellin transport dataset as a separate study with a focus separate from sugar transport.

3. Han et al. have shown a dimer unit transport model. Can the authors comment on why they have considered to focus on the monomer model instead the dimer model?

SWEET, and the bacterial homologue SemiSWEET, transporters have been simulated in other studies besides this preprint:

Latorraca, N. R., Fastman, N. M., Venkatakrishnan, A. J., Frommer, W. B., Dror, R. O. & Feng, L. Mechanism of substrate translocation in an alternating access transporter. *Cell.* **169**, 96-107. (2017) <https://doi.org/10.1016/j.cell.2017.03.010>

Jia, B., Zhu, X. F., Pu, Z. J., Duan, Y. X., Hao, L. J., Zhang, J., Chen, L.-Q., Jeon, C. O. & Xuan, Y. H. Integrative view of the diversity and evolution of SWEET and SemiSWEET sugar transporters. *Front. Plant Sci.* **8**, 2178. (2017) <https://doi.org/10.3389/fpls.2017.02178>

Selvam, B., Yu, Y.-C., Chen, L.-Q. & Shukla, D. Molecular basis of the glucose transport mechanism in plants. *ACS Cent. Sci.* **5**, 1085-1096. (2019) <https://doi.org/10.1021/acscentsci.9b00252>

Cheng, K. J., Selvam, B. Chen, L.-Q. & Shukla, D. Distinct substrate transport mechanism identified in homologous sugar transporters. *J. Phys. Chem. B.* **123**, 8411-8418. (2019) <https://doi.org/10.1021/acs.jpcc.9b08257>

Deng, Z., Yuan, B. & Yang, K. Cardiolipin selectively binds to the interface of VsSemiSWEET and regulates its dimerization. *J. Phys. Chem. Lett.* **12**, 1940-1946. (2021) <https://doi.org/10.1021/acs.jpcclett.1c00022>

Weigle, A. T., Carr, M. & Shukla, D. Impact of increased membrane realism on conformational sampling of proteins. *J. Chem. Theory Comput.* **17**, 5342-5357. (2021) <https://doi.org/10.1021/acs.jctc.1c00276>

Isoda, R., Palmai, Z., Yoshinari, A., Chen, L.-Q., Tama, F., Frommer, W. B. & Nakamura, M. SWEET13 transport of sucrose, but not gibberellin, restores male fertility in *Arabidopsis sweet13;14*. *Proc. Natl. Acad. Sci. U. S. A.* **119**, e2207558119. (2022) <https://doi.org/10.1073/pnas.2207558119>

Fatima, U., Balasubramaniam, D., Khan, W. A., Kandpal, M., Vadassery, J., Arockiasamy, A. & Senthil-Kumar, M. AtSWEET11 and AtSWEET12 transporters function in tandem to modulate sugar flux in plants. *Plant Direct.* **7**, e481. (2023) <https://doi.org/10.1002/pld3.481>

In each of these studies, only monomeric SWEET transporter models are used. This could be because SemiSWEET, which lacks transmembrane helix 4, forms a parallel homodimer that is of a relatively similar size as a SWEET monomer. Since a SemiSWEET crystal structure was resolved first, published SemiSWEET MD simulations preceded SWEET simulations. This could have created a precedent for the simulation of this transporter family that SWEET monomer simulation is an acceptable practice. Additionally, the simulation of just a SWEET monomer is more computationally efficient, as the atom count is less in this case.

We have proceeded with simulations of monomer AtSWEET13. That being said, we do agree with Reviewer 2 and believe there is importance to representing SWEET transporter oligomerization in modeling and simulation studies. To this end, we have an additional simulation study where we are studying the dynamics of the homotrimer structure (PDB ID: 5CTH) of OsSWEET2b. The results of this future work are not part of this manuscript. This project also has required significant computational time and resources as well.

C. The authors should organize their data and find better ways to present and summarize their data in a concise and elegant fashion. There are 40 supplementary figures with multiple subpanels. Their supplementary file looks like a “data dump”. Similar to the figures the text is also very lengthy and needs major improvement. Specific questions regarding the presentation of the data:

- 1. What is the point of including bar plots for the tIC-correlation features (Figs. S15-S17) where all the bars show high values? Is the correlation value for each bar significant to the story? What is the intended message of all of these bar plots?**

Original Figs. S15-S17 described the molecular descriptors used as input for tICA decomposition and their correlation with either tIC1 or tIC2 for each respective system. The purpose of showing all these plots was for the sake of transparency when we claim what processes are associated with tIC1 versus tIC2 for AtSWEET13 *apo*, *holo* SUC, and *holo* GLC transport processes. From our initial “complex” tICA, there were many features used as input.

Here, we have shortlisted the set to show which ones were actually correlated with each of the tICs. In the main text we list the following results:

- tIC1^{apo} signified protein conformational change
 - o (Figure S13 only consisted of protein-protein distances)
- tIC1^{SUC} reflected glucosyl moiety atom positioning
 - o (Figure S15 consisted of protein-SUC glucosyl moiety atom distances, along with some protein-protein distances)
- tIC2^{SUC} reflected fructosyl moiety atom positioning
 - o (Figure S17 only consisted of protein-SUC fructosyl moiety atom distances)
- tIC1^{GLC} reflected protein conformational change
 - o (Figure S14 only consisted of protein-protein distances)
- tIC2^{GLC} reflected monosaccharide substrate atom positioning
 - o (Figure S16 only consisted of protein-GLC atom distances)

The purpose of the data allows the reader to understand the rank importance of each feature based on the strength of their correlation to the tIC. To maximize transparency and demonstrate rigor of our analyses, these data were shown in addition to the feature-diverse input sets in Tables S3-S5. To help with interpretability, we have reorganized this data. First, we removed the lengthy Tables S3-S5. Instead, we now include the exact tICA correlation values from those graphs into new Supplementary Tables 3-7.

2. **The authors have projected RMSD values onto the protein structure in Figs. 2 and S9. It is very difficult to understand the significance of these images as the colors of the most important regions are very close to one another. I was not able to confirm the claim that “the average per-residue RMSD is greater for overall GLC transport than for SUC” (p. 6), as the RMSD of terminal residues are the highlight of these figures, which is not of interest. Instead of projecting the values on the structures of these figures, the authors should use RMSD plots to evaluate whether there is a significant difference between glucose and sucrose. They should specify which regions show the greatest difference and most importantly, assess whether this difference is significant.**

Our statement concerning how “the average per-residue RMSD is greater for overall GLC transport than for SUC” was meant in reference to the original Figure S9. We agree with Reviewer #2 that our Main Text presentation is confusing in that Figure 2 does not show structures which directly communicate this information. Additionally, the terminal residues do demonstrate the largest per-residue RMSD out of the entire protein.

To avoid over-generalization and to address Reviewer #2’s comments about changes in RMSD, we made the following changes:

- (1) We first changed Main Text Figure 2c and 2d by just showing the worms representation of the states undergoing conformational change associated with alternate access. RMSD values calculated from alternate access implicate the greatest extent of

conformational change throughout the AtSWEET13 transporter during both GLC and SUC transport.

- (2) We present a plot showing the RMSD as a function of protein residues for the conformations related to alternate access in Figure 2e and 2f. We tried many different forms of presentation for this analysis and found that the best approach to show which parts of the protein move differently can be seen during commitment to alternate access. Alternate access is the stage of transport which invokes the greatest extent of protein conformational change.

- (3) The new Figure 2 appears as follows:

Figure 2. Protein conformational change implicated in sugar transport. (a) GLC transport versus the difference in intracellular versus extracellular gating. AtSWEET13 is in a more IF-like position the more positive the gating difference and a more OF-like position the more negative the gating difference. (b) SUC transport versus the difference in intracellular versus extracellular gating. (c) AtSWEET13 OF-HG-like conformation immediately following commitment to alternate access in GLC transport. (d) AtSWEET13 OF-HG-like conformation immediately following commitment to alternate access in SUC transport. (e) RMSD values relating to residue-specific fluctuations in response to alternate access during GLC transport. (f) RMSD values

relating to residue-specific fluctuations in response to alternate access during SUC transport. Arrows drawn over landscapes are for illustrative purposes to suggest the transport path according to the lowest MSM-weighted free energy transitions as shown. RMSD values are calculated as an average in comparison to the number of states found at the center of each energetic minima characterizing each enumerated state in panels **(a)** and **(b)**.

- (4) We removed the sentence concerning greater average per-residue RMSD so that we can focus instead on individual residues with high RMSD fluctuations. We further describe these results in a new paragraph, talking about the specific residues and at what stage during the transport cycle do these high fluctuations occur.
- (5) We introduce a new Figure 3 to further dissect the RMSD analysis results and illustrate the new findings from the data. Here we show the new Figure 3:

Figure 3. Protein conformational change implicated in sugar transport. (a) RMSD fluctuations throughout GLC transport that are statistically significantly different from RMSD fluctuations seen during AtSWEET13 alternate access for SUC transport. (b) RMSD fluctuations throughout SUC transport that are statistically significantly different from RMSD fluctuations seen during AtSWEET13 alternate access for GLC transport. The colorbar represents the number of states for which the RMSD values significantly differ from that of the alternate access state of the opposite ligand transport process. HG-like states undergoing alternate access are shown as snapshots for GLC (left) and SUC (right) transport, respectively. Residues with heatmap entries marked with an asterisk showed statistical significance at $p < 0.01$ by t-test. Residues without an asterisk showed statistical significance at $p < 0.05$.

These changes have resulted in an extension of Results Section 2. Results Section 2 has been renamed from “Conformational change cannot explain the differences in AtSWEET13 sugar transport” to “Alternate access for selective sugar transport is distinguished by altered energy barriers and distinct residue fluctuations”.

To better transition between these RMSD and our atom-specific analyses, we have distinguished our tICA decomposition results as a separate section, titled “Evaluating the contribution of conformational change versus ligand recognition to explain differences in AtSWEET13 sugar transport”.

We now go into detail about the specific residues that are implicated with alternate access. We recommend Reviewer #2 see the added results, but we now conclude this section with the following sentence:

“In total, these RMSD analyses help explain what residue-specific dynamics deviate throughout conformational change associated with GLC versus SUC transport events.”

- 3. In Figs. 2A and 2B, the authors show plots linking the difference in intracellular versus extracellular gating to the sugar position. They also plot this for other atoms of the sugar molecules in Figs. S3-S8. What is the reasoning behind choosing glucose O5 and sucrose O3 atoms to represent all of these plots in the main figure? The only reason given by the authors was that “they represent the same ether oxygen along nearly identical hexose units” (p.10). Considering that there are other oxygens in identical positions in glucose and sucrose, why was this oxygen chosen? Similarly, what is the reasoning behind choosing the structures in Fig. 2C and 2D among all the other ones in Fig. S9? Similarly, what is the reasoning behind choosing the panels in Fig. 3B among all the other ones in Fig. S18-S35?**

With respect to choosing the AtSWEET13 structures shown in Figure 2C and 2D, we are instead substituting them with just the states relating to alternate access, as these states show the most distinct conformational change.

We apologize for any further confusion caused and would like to elaborate why certain atoms were selected and presented in the main text.

When comparing GLC versus SUC in landscapes, our strategy was to select atoms which would appear to be most comparable between the two sugar substrates.

There are some cases where one would immediately assume that some atoms in SUC are not structurally and chemically equivalent to GLC. The ether linkage connecting the hexose and fructose units in SUC causes a chiral inversion at the glycosidic linkage site; as such, these oxygens and their parent backbone carbons are not chemically equivalent. Similarly, GLC does not possess any pentose moieties seen in SUC. That is, GLC is not the same molecule as fructose. Therefore, these atoms are not chemically equivalent either. Likewise, because GLC does not contain a pentose moiety, the centers of mass for GLC and SUC would not be equivalent either.

As we performed all our atom-specific transport and rotation analyses, we synthesized the information contained within all our landscapes. We came up with what we believed were fair and ethical ways of comparing the two molecules, and succinctly presenting our findings in the Main Text.

For explaining differences in alternate access and transport, our efforts culminated in showing the hexose backbone ether oxygen transport landscapes in Figure 2. That is because our initial guess was that these atoms should be structurally and functionally equivalent between GLC and SUC molecules. However, the free energy landscapes show that these atoms experience different barriers to transport.

When comparing how positional rotation (θ_{xy}) varies between the sugars, we selected the GLC C3 and SUC C5 atoms. These atoms correspond to a backbone hexose carbon whose corresponding alcohol bears the same chirality whether incorporated into monomeric GLC or disaccharide SUC. These atoms also had some of the greatest similarity in their landscapes across the two molecules. With respect to the θ_{xy} positional rotation analyses, the landscapes shown in the SI for the corresponding alcohols present the exact same trends. Aside from this selection, there are some general trends that are specific to atoms from each molecule. GLC atoms have a very constrained intracellular binding path, which we had already discussed in the text. Meanwhile, some SUC atoms can adopt the GLC intracellular binding pathway, although they more readily sample parts of the AtSWEET13 transmembrane channel which GLC atoms do not. This point was discussed in the initial submission of this manuscript.

It is worth noting that literature on sugar transporter structure-activity relationships endeavors to find which alcohols are most critical for sugar binding and transport. We further elaborate upon our identification of GLC C3/O3 and SUC C5/O5 hydroxyls as critical by relating this finding to literature on GLUT1, GLUT2, GLUT3, and PfHT1 sugar transporters.

When comparing the facial inversion differences between the sugars, the exact choice of atoms did not exactly matter when comparing θ_{xz} or θ_{yz} analyses. From each atom-specific inversion landscape, it can be seen that atoms within the monosaccharide GLC experience free facial inversion throughout transport (although at a high barrier). In contrast, the SUC atoms realistically experience only a single inversion that must occur during commitment to alternate access (x-axis Z coordinate transitions from 0 \rightarrow 5 Å). To this end, we again chose the hexose backbone ether oxygen atoms, GLC O5 and SUC O3, to highlight the differences in facial inversion between substrates.

Our original Figure S36 (now Supplementary Figure 30) is meant to highlight the functional groups considered critical for molecular recognition by AtSWEET13. We thank Reviewer #2 for their comment because this means that our logic was not readily available for some readers to interpret and appreciate given the way the text was written. We have now provided explicit rationale.

We have revamped this section of the text to make these findings more clear, as mentioned in response to *Comment A.2*.

- Following up on point 3, instead of producing many plots that essentially show the same thing (for example the plots they produced using the z-coordinates of each sugar atom in Figs. S3-S8) the authors should use more comprehensive metrics that can illustrate their finding (e.g. using the z-coordinate for the center of mass of glucose).

Here we show what the z-coordinate transport landscapes look like when considering the center of mass (COM) for either GLC, SUC, the hexose subunit of SUC, or the pentose subunit of SUC:

Response Letter Figure 1. COM z-coordinate transport landscapes for (a) GLC, (b), entire SUC, (c) hexose subunit of SUC, and (d) pentose subunit of SUC.

As can be seen, the landscape matches well with the data we have shown in the Main Text. That is, we had originally selected landscapes to show which we felt were representative of the data seen on a per-atom basis.

When we first began this study, our transport landscapes (like those shown in Figure 2A and 2B) were initially plotted using a central atom of SUC or GLC. This central atom was used as a proxy for the center of mass. As we proceeded with our analyses, we realized that selection for different atoms revealed different free energy landscapes in events leading to and through alternate access. This result became most apparent when we compared the transport landscapes in Figures S3-S8, where the fructosyl atoms on SUC possessed a

lower free energy barrier for alternate access, but the hexose atoms on SUC did not. In fact, the hexose atoms on SUC, which structurally correspond to atoms on GLC, maintained an elevated barrier for alternate access. The exact trend could be observed when comparing monosaccharide GLC versus disaccharide SUC transport, as shown in Main Text Figure 2. The same result can be seen here when comparing **Response Letter Figure 1c and 1d**. However, when the COM for all of SUC is used, this information becomes lost.

Ultimately, landscape representation becomes a preference. We agree that the use of COM can help simplify arguments. Our ability to identify differences between GLC and SUC transport stemmed from an early realization that different parts of SUC are perceived differently, which is something we would like to preserve in the manuscript. While the z-coordinate transport landscapes could be re-represented using the COM, the theta rotation analyses cannot due to how they are calculated. We have clarified our rationale behind selecting which atomic landscapes to represent in the Main Text (see *Comment C.3*).

- 5. Most of the data is presented in heat maps (Figs. 1B, 2A, 2B, 3B, S1-S8, S18-S35, S37-S39). The heat maps in the main figures contain numbers and arrows, but it is not clear what these numbers mean or which states the numbers correspond to. The heat maps in the supplementary do not contain any labels and it is unclear what the reader should take away from all of these plots. Such plots should be accompanied by representative snapshots. Again, better metrics should be defined to present the results in a concise way and different modes of data visualization techniques should be used to facilitate understanding (e.g., chord diagrams for free energy where each state represents a node, with conformational snapshots linked to each node).**

With respect to the gating transport plots shown in Figures S1-S8, we show representative conformations/snapshots in Figure S9.

For the atom-specific plots concerning rotation and inversion of substrate atoms, our original intent was to simply show that the free energy landscapes differed by selecting different atoms as references for monitoring sugar transport. To this end, we were able to point out overarching trends between what could be identified as individual motifs. While we have not labeled each landscape, general metastable states can be made out with reference to Main Text figures. Again, we provide the data to maintain transparency with the reader so that our claims of general trends in atom-specific recognition can be supported.

- 6. Figure 4 is very difficult to understand as the transporter is shown from different angles in the snapshots. The authors should improve this to show the transporter from the same side view angle in each snapshot and use slab representations to show the different states and ligand association/dissociation. If needed, they should use zoomed-in insets to show the ligand orientation.**

We thank Reviewer #2 for this spectacular suggestion, as we feel the revised Figure 5 looks quite sharp. We took heavy inspiration from the recently published work of Birgit Schiøtt's group on the SUC1 plant sucrose transporter (see Bavnhoj *et al.* (2023) *Nature Plants*,

<https://doi.org/10.1038%2Fs41477-023-01421-0>). We give credit to this work in the Figure 5 caption:

Figure 5. Proposed mechanism of substrate recognition in AtSWEET13 transport of GLC (a) and SUC (b). Substrate inversion happens freely throughout GLC transport, making facial flipping not necessarily required for GLC to pass through the Trp58-Trp180 binding pocket during HG conformations and commitment to alternate access. For SUC transport, substrate inversion does not happen freely throughout the transport process, requiring significant substrate conformational rearrangement before AtSWEET13 can commit to alternate access. Figure inspired by the proposed mechanism for substrate binding in SUC1.⁴²

Again, we are very thankful that Reviewer #2 made this fantastic suggestion, as we feel it improves the interpretability of the manuscript. As an alternative view of the transport event, we moved the old Figure 4 to SI, now as Supplementary Figure 31.

7. **The second section of Results is titled “Conformational change cannot explain the differences in AtSWEET13 sugar transport”. The authors should not focus on the aspects that cannot explain the differences in sugar transport, but instead concisely explain which aspects can.**

See response to Comment C.2.

We have renamed this Results Section from “**Conformational change cannot explain the differences in AtSWEET13 sugar transport**” to “**Alternate access for selective sugar transport is distinguished by altered energy barriers and distinct residue fluctuations**”.

8. **The text needs to be revised to remove the sections about the details of the methodology. Only brief descriptions of the methodology should be included in the Results section, and these descriptions should be understandable by the broader audience of the journal.**

Our submitted draft of this manuscript has been revised to talk about tICA decomposition in a newly labeled results section, “**Evaluating the contribution of conformational change versus ligand recognition to explain differences in AtSWEET13 sugar transport**”. Of the 734 words in this section, 125 words (~17% out of the Section) are dedicated to explaining some background behind deciding features for discretizing the simulation data using tICA. The exact methodology is explained in greater detail under the Methods subsection “Determination of agnostic MSM descriptors”. Considering the broad audience of the *Communications Biology* journal, an explanation is needed before discussing tICA. The tICA decomposition, while a preceding method to the MSM workflow, is a result in and of itself.

We feel that our current description of how tICA is used to differentiate contributing factors to transport is not overly technical. Discussion on how MSMs are constructed and how featurization equates to structurally, dynamically, and biologically significant findings has also been reported in journals with just as broad of an audience as *Communications Biology* (see Unarta *et al.* ... & Huang (2021) *PNAS*, <https://doi.org/10.1073/pnas.2024324118>).

D. The rigor of the simulations should be improved as well. Specific concerns regarding the rigor of simulations:

1. **At what pH were the protonation states determined? What was the protonation state of D189? How were the histidines modelled (protonated or neutral, and if neutral, in which tautomeric state (delta/epsilon))?**

With respect to pH

Protonation states were determined at a pH of 5.3 and were determined using the PROPKA webserver. D189 was modeled in the anionic form because the pH reference for

simulations was 5.3. D189 is positioned along the apoplastic, or extracellular, face of the plasma membrane.

Selecting a pH of 5.3 was chosen because the apoplastic side of the plant plasma membranes is known to be more acidic than the cytoplasmic side (which is between 7-7.5).

With respect to histidines

The AtSWEET13 crystal structure (PDB ID: 5XPD) resolves an inward-facing transporter that is truncated to residue 222. Accordingly, the simulations were performed with an AtSWEET13 transporter comprised of residues 1 to 222. In reference to the UniProt sequence of AtSWEET13 (Accession ID: Q9FGQ2), there are no histidines present in the truncated sequence. Therefore, no histidines are present in our model AtSWEET13 structure for protonation or tautomeric state to be considered.

Accordingly, the rigor of our AtSWEET13 simulations is maintained by recognizing the need for modeling an appropriate pH and membrane environment. We include the explicit mentioning of the pH we used to model our simulations under the section “***System parameterization.***”

We include:

“The pH was determined to be 5.3 to replicate the apoplastic side of the plasma membrane in a plant cell.”

Evaluating simulation setup rigor

Our simulation setup takes recognizes the more acidic apoplastic side of the plant plasma membrane. Additionally, we make use of a realistic plant plasma membrane bilayer. Often times, the default mantra of performing MD simulations of membrane proteins with homogeneous POPC bilayers is extended to plant membrane proteins, despite POPC being recognized as one of the least abundant lipid species present in plant plasma membrane bilayers (see Weigle, Carr, and Shukla (2021) *J. Chem. Theory Comput.*, <https://doi.org/10.1021/acs.jctc.1c00276>). Lastly, our simulations were performed using a Langevin thermostat and Monte Carlo barostat.

- 2. Figs. 1, 2, and 4 include conformational snapshots from simulations. How were these conformational snapshots selected? Can the authors show tests to ensure that these are representative?**

As was described in our original submission under the Methods subsection titled, “***Protein structure visualization***”:

“Structures selected for Main Text and Supplementary figures were chosen from the centers of different metastable minima along relevant free energy landscapes.”

We provide justification for the choice of our atomistic landscapes under *Comment C.3.*

Having selected our snapshots nearest to the center of each free energy minima, these snapshots will represent some of the most stable states within the free energy landscape surface. Candidate frames were extracted from the landscapes by using a sort of coordinate search. For example, a pseudo-code representation would look like this:

```
- suppose X1 and X2 are designated x-axis values along
a free energy landscape
- suppose Y1 and Y2 are designated y-axis values along
a free energy landscape

- provided the landscape, draw a box enclosed around
the most energetically stable center of a metastable
free energy minima

X1 < desirable_snapshot_x_value < X2
Y1 < desirable_snapshot_y_value < Y2

for each frame in each trajectory:
    if (X1 < frame_x_value < X2)
        AND
        if (Y1 < frame_y_value < Y2):
            extract frame
```

Once all the extracted frames were collected, the representative frames chosen for the manuscript were usually taken from the middle of the selection. The logic here was that since these frames were in the middle of the selection, they should be representative of frames whose data points reside in the middle of the (X1,Y1), (X2,Y2) distribution.

3. Which form of glucose was modelled (D/L glucose and alpha/beta-glucose)? Was this the most relevant form for this transporter?

Beta-D-glucose was used for modeling and simulation. Beta-D-glucose is the primary form of glucose transported by AtSWEET13. In experimental studies measuring the glucose transporting capacities of AtSWEET13, such as in the *Han et al. (2017) PNAS* paper, FRET sensor sugar uptake assays are used. The FRET sensor used for monitoring glucose transport is specifically tuned for D-glucose:

Ref. 34 of *Han et al. (2017) PNAS*;

see Hou, B.-H., Takanaga, H., Grossman, G., Chen, L.-Q., Qu, X.-Q., Jones, A.-M. Lalonde, S., Schweissgut, O., Wiechert, W. & Frommer, W. *Nat. Protoc.* **6**, 1818-1833. (2011)
<https://doi.org/10.1038/nprot.2011.392>

Despite this sensor being used to measure glucose transport by AtSWEET13, this sensor was originally developed to measure glucose levels in mammalian cells. The natural form of mammalian glucose is beta-D-glucose. Considering that experiments done on AtSWEET13

glucose transport are done measuring beta-D-glucose, the use of beta-D-glucose in these simulations is justified.

- 4. The authors states that “All holo AtSWEET13 systems were built using 100 mM of substrate to match published experimental conditions.” (p. 20). Can the authors point to the location where in the referenced article this is mentioned? Can the authors mention how many sugar molecules were in the simulation box?**

We thank the reviewer for catching this typo and sloppy, but honest, mistake in the Methods section of our text. Referring back to our original notes when constructing the systems, any reference for “published experimental conditions” was originally intended for referencing what ions were to be used for the simulations, not the sugar concentration. We have added the citation to support our use of KCl for neutralization of the systems. With respect to sugar concentration, we selected value of 100 mM to approximate the general range of sugars present within a phloem cell.

Again, we apologize for this error and thank Reviewer #2 from preventing this mistake.

We have corrected this error as follows:

All AtSWEET13 systems were neutralized using KCl to match published experimental conditions.⁵⁹ Sugar concentrations were modeled as 100 mM to approximate the sugar concentration of phloem cells as it is reported in plant physiology textbook sources.^{60,61} Because of the established sugar concentration, 24 individual sugar molecules were packed into the box.

[59]. Peuke, A. D. Correlations in concentrations, xylem and phloem flows, and partitioning of elements and ions in intact plants. A summary and statistical re-evaluation of modelling experiments in *Ricinus communis*. *J. Exp. Bot.* **61**, 635–655 (2010).

[60]. Hall, S. M. & Baker, D. A. The chemical composition of *Ricinus* phloem exudate. *Planta* **106**, 131–140 (1972).

[61]. Buchanan, B. B., Gruissem, W. & Jones, R. L. *Biochemistry & Molecular Biology of Plants*. (American Society of Plant Physiologists, 2000).

- 5. How were the lag times determined? Which tests were performed for model validation?**

We thank Reviewer #2 for this question, as this means that this information was not readily available in the main text. First, we will better describe how we constructed our Markov state Models (MSMs).

In the methods section of the text under “*Markov state model (MSM) construction*”, we address how we make use of rigorous state decomposition using tICA to ensure that our extent of sampling partially addresses concerns of ergodicity and detailed balance. During the MSM construction protocol, we performed an initial pilot run to determine

what the range of search values should be for cluster count and tIC dimensionality. For example, our initial cluster count grid search spans either 100, 200, 300, 400, 500, 750 and then 1000 clusters. Our initial tIC dimensionality grid search spans either 2, 5, 9, and 12 tICs. During this initial grid search, we make a guess as to what should be our initial lag time. For this early pilot run, we assumed 40 ns as our lag time because it was half of our 80 ns trajectory time.

Based upon how the implied timescale plots appeared for these initial MSM pilot runs, we selected a “working” lag time equivalent to the shortest trajectory timestamp for which we observed a converged implied timescale. By selecting the shortest possible Markovian lag time in this manner, we can optimize the number of jumps that we observe and maximize the resolution of the MSM processes.

We then committed the “working lag time” to our MSM construction protocol. If our optimized MSM did not have a converged implied timescale plot at the specific lag time, then the “working” lag time was reselected and the optimization protocol was restarted.

With respect to model validation, we failed to mention use of the Chapman-Kolmogorov (C-K) test. The C-K test evaluates whether the Markov state model achieves detailed balance at a given lag time, thereby validating whether the global flow of probability is conserved for a given MSM at its given lag time.

Also mentioned in this section of the text, we validate our MSMs by ensuring that our tICA landscapes are connected (i.e., introducing a dataset obeying assumptions of ergodicity, dataset has reversible sampling). Text was added to the methods section to better describe how this check is a form of validation.

After making the MSM, we also evaluate the extent of reweighting raw cluster counts probability versus MSM-weighted cluster counts probability. This validation test ensures that our choice of molecular descriptors used to make the MSM preserve the underlying phase space that was captured during adaptive sampling. If MSM-reweighting of free energy landscapes otherwise became exaggerated, then the metastable states on the free energy landscapes would be misrepresented. Without this validation test, presentation of states occupying the free energy minima would be less trustworthy. Text was added to the methods section to better describe how this check is a form of validation.

What we had forgotten in the previous submission, we now also include the Chapman-Kolmogorov (C-K) test to ensure that Markovian assumptions of detailed balance are satisfied when constructing an MSM with the working lag time.

We have further elaborated on our protocol in the main text:

“PyEMMA software (v2.5.6) was used for MSM construction.⁹² With exhaustive, agnostic features determined, MSMs were generated using an iterative grid search procedure where the VAMP score was optimized.⁹³ The MSM protocol used herein can be described as follows:

- (1) Given the exhaustive set of agnostic features, construct pilot MSMs to determine the range of cluster counts and time-Independent Components (tICs) capable of returning converged implied timescale plots.⁹⁴ From these pilot runs, obtain a working MSM lag time to use for a more exhaustive grid search based on visual inspection of resulting implied timescale plots. A “working” lag time is one that is equivalent to the shortest trajectory timestamp for which a converged implied timescale was observed. By selecting the shortest possible Markovian lag time in this manner, the number of observed jumps within individual trajectories is optimized and the resolution of MSM processes is maximized.
- (2) Perform an exhaustive grid search where the accepted hyperparameter combination optimizes the VAMP score. With this initial MSM transition probability matrix determined, project tIC1 versus tIC2 from the tIC analysis (tICA) object output during clustering.
- (3) Confirm that the tICA decomposition landscape from Step 2 is well-connected. Having a well-connected landscape can be indicative of reversible sampling, which is necessary to maintain the Markovian assumption of detailed balance. If the resulting tICA decomposition landscape is well-connected, then determine which features are most correlated with tIC1 and tIC2 for the currently optimized MSM. If not, revisit adaptive sampling to connect this complex tICA landscape with more data.
- (4) Once the tICA landscape has been connected, plot the raw cluster counts versus the MSM-weighted cluster counts for the tICA-converged dataset. This validation test ensures that the choice of molecular descriptors used to make the MSM preserves the underlying phase space. Otherwise, resulting free energies may appear exaggerated and result in misrepresentation of metastable states. If the extent of reweighting between the raw and MSM-adjusted data deviates beyond a relationship of $x=y$ where y is much greater than x , then the molecular descriptors used for MSM featurization are deriving an overtuned bias correction (Supplementary Figure 36).^{95,96} In this case, molecular descriptors that are highly correlated to tIC1 and tIC2 of the exhaustive tICA decomposition can be used to simplify the MSM featurization step.
- (5) Repeat Steps 1-3 using molecular descriptor feature sets inspired from Step 4 until the raw cluster counts versus MSM-reweighting plots approximately satisfy a relationship of $x=y$.
- (6) Perform the Chapman-Kolmogorov (C-K) test to ensure that Markovian assumptions of detailed balance are satisfied when constructing an MSM with the working lag time (Supplementary Figures 37-40).

The final feature sets for each system are listed in Supplementary Table 1. MSM hyperparameters used for each system are described as follows. *Apo* MSM hyperparameters: 950 clusters, 3 tICs, a tICA lag time of 8 ns, and an MSM lag time of 30 ns. GLC MSM hyperparameters: 750 clusters, 12 tICs, a tICA lag time of 8 ns, and an MSM lag time of 20 ns. SUC MSM hyperparameters: 900 clusters, 8 tICs, a tICA lag time of 8 ns, and an MSM lag time of 40 ns (see Supplementary Table 2).”

- 6. The authors have mentioned seeded trajectories were simulated for at least 80 ns in length, depending on available computational resources.” (p. 22) How long were the simulations?**

MSMs were adapted to molecular dynamics simulations as an enhanced sampling technique based on the realization that (1) not all researchers have equal access to powerful computers capable of capturing long timescale events within singular simulations, and (2) that the stitching together of separate unbiased trajectories could still preserve ensemble thermodynamic and kinetic properties when combined under an MSM {see Bowman, G. R., Ensign, D. L. & Pande, V. S. Enhanced modeling via network theory: Adaptive sampling of Markov state models. *J. Chem. Theory Comput.* **6**, 787-794. (2010) <https://doi.org/10.1021/ct900620b>}

As stated in the acknowledgements, this research was performed using the Blue Waters sustained-petascale computing project. Across the Kepler K20X GPUs installed throughout the distributed computing made available on Blue Waters, our simulations could achieve an average efficiency of 40 ns/day. Given that all Blue Waters supercomputer user wallclock times were capped at 48 hours, at most our simulations could be run for 80 ns. However, given available computational resources, or during periods where Blue Waters temporarily extended their wallclock times for testing purposes, instances could potentially arise where a round of simulation could also be run with longer individual trajectories. The exception for which we made this reference in the text was for one of our *apo* AtSWEET13 rounds, where we were able to conduct using trajectories with a length set to 280 ns for that single round.

It has been shown that using variable lengths of simulated trajectories when constructing a Markov state model does not compromise the quality of the resulting model so long as trajectories are only used which are at least as long as the MSM lag time {see Figure S23; Shukla, D., Meng, Y., Roux, B. & Pande, V. S. Activation pathway of Src kinase reveals intermediate states as targets for drug design. *Nat. Commun.* **5**, 3397. (2014) <https://doi.org/10.1038/ncomms4397>}. In the case of our *apo* AtSWEET13 simulations, our MSM lag time was found to be 30 ns. Whether the trajectories were 80 ns in length or 280 ns in length, they would each record at least one Markov jump occurring with a lag time of 30 ns.

To this end, the rigor of our study is not compromised because some trajectories are longer than others. All trajectories used do satisfy the assumptions of Markovianity needed for MSM construction and validation.

E. Minor comments:

- 1. In Fig. 1, the authors show the *apo* AtSWEET13 transport cycle. It would be more compelling if the authors combine the holo and *apo* transport cycles showing ligand association/dissociation steps. In essence, Fig. 1A and Fig. 4 show the same cycle and it is confusing why the cycle is shown multiple times.**

One of the goals of our manuscript was to distinguish how the different sugar substrates are perceived by AtSWEET13 during their respective transport cycles.

We therefore outlined our manuscript to first describe the general transport cycle, and to then monitor ligand transport.

The flow of the manuscript is as follows:

- In Figure 1A. We show that we have resolved the complete transport cycle of AtSWEET13 in the *apo* form as well as in the *holo* forms when introduced to sucrose (SUC) or glucose (GLC). Further justification for why Figure 1 is formatted the way it is:
 - o Figure 1A introduces the transmembrane helices, important residues, and representative along with the transport cycle
 - o All distinct conformational states observed in the *apo* conformational cycle here are also observed in the *holo* transport cycles. Therefore, we show the *apo* case to be representative
- We then proceed to describe the transport cycle with respect to the orientation of the ligand, as our major finding is on how the substrates are differentially perceived and translocated by AtSWEET13. Figure 4 then actually depicts the specific discriminative sugar transport events described by sugar placement. These discriminative sugar placement events reflect differences in how either sugar is transported, rather than differences in the transporter protein conformation itself.

To remedy these concerns and reflect our intentions, we now deliberately elaborate the exact purpose of each section.

2. There are many instances in the text where the authors have referred to the wrong supplementary figure, which should be fixed.

We have reviewed our referencing of Supplementary Figures. We have corrected our Supplementary Figure references, as well as made these references more specific and deliberate within the Main Text.

3. By convention, the chair form of a sugar is drawn with the ring oxygen at the back right position. The authors should use the conventional chair notation in their figures, or explain their reasoning if they choose not to.

We drew our sugars in a nonconventional style to reflect how they appear when modeled in three dimensions. We wanted to draw the sugar structures in a three-dimensional orientation so that any structure-activity relationships we derive can be more easily related to our future work depicting gibberellin transport. When drawn using conventional chair notation, it is difficult to perceive much structural similarity between the sugar substrates versus gibberellin.

For example, if we extend the structure-activity relationships (SAR) that we had deduced from Figure S36 and compare them to our unpublished data on AtSWEET13 gibberellin GA3 transport, our drawing style makes the sugar SAR more comparable to the GA3 chemical structure.

Beta-D-Glucose (GLC)

Sucrose (SUC)

Gibberellic Acid (GA3)

REVIEWERS' COMMENTS:

Reviewer #1 (Remarks to the Author):

The authors fully addressed all of my comments from the previous round of reviews. The level of detail in their responses, as well as the addition of a new subsection dedicated to the issue of the interaction mechanism between sugars and AtSWEET13, significantly surpasses the typical standard for revisions. In my opinion, the paper can be published in its current form.

Reviewer #3 (Remarks to the Author):

I have been requested to simply provide feedback on how suitably Reviewer #2's comments have been addressed, and if the manuscript is therefore suitable for publication.

After reading the manuscript and rebuttal letter, I conclude that the raised issues were addressed and that the proper amendments were made.

Despite Reviewer #2's critical stance on this manuscript, I think it is ready to be published and further scrutinized by the scientific community.

During peer review, we received comments from one of the reviewers concerning the chemical representation of our sugar molecules. After formatting a separate manuscript for Nature Communications, we made the decision to correctly redraw the chemical structures for glucose and sucrose using the Nature Publishing Group's ChemDraw template settings. All figures containing sugar chemical structures have been updated.

As we were preparing our Reporting Statistics, we made slight corrections to Main Text Figures 3 and 6. In Main Text Figure 3, we reproduced our statistical analyses and found that there was a slight miscalculation in the previous version. Some of the residues highlighted as significantly different between distinct transport stages were not as significant across as many transport stages. These changes do not affect any of the text in the related Results section of the manuscript, but just the appearance of Main Text Figure 3. For our original Main Text Figure 6, we had reported statistics after removal of outliers, but had presented data distributions where the means and quartile ranges were calculated with outliers still included. The current distributions shown in Main Text Figure 6 now have outliers removed.